# Discrepancies in the Simulated Global Terrestrial Latent Heat Flux from GLASS and MERRA-2 Surface Net Radiation Products

**Xiaozheng Guo [1], Yunjun Yao [1,\*], Yuhu Zhang [2], Yi Lin [3], Bo Jiang [1], Kun Jia [1], Xiaotong Zhang [1], Xianhong Xie [1], Lilin Zhang [4], Ke Shang [1], Junming Yang [1] and Xiangyi Bei [1]**

[1] State Key Laboratory of Remote Sensing Science, Faculty of Geographical Science, Beijing Normal University, Beijing 100875, China; boyxiaozheng@mail.bnu.edu.cn (X.G.); bojiang@bnu.edu.cn (B.J.); jiakun@bnu.edu.cn (K.J.); xtngzhang@bnu.edu.cn (X.Z.); xianhong@bnu.edu.cn (X.X.); shangke@mail.bnu.edu.cn (K.S.); julming@mail.bnu.edu.cn (J.Y.); xiangyibei@mail.bnu.edu.cn (X.B.)

[2] College of Resource Environment and Tourism, Capital Normal University, Beijing 100048, China; yuhu.zhang@cnu.edu.cn

[3] School of Earth and Space Sciences, Peking University, Beijing 100871, China; yi.lin@pku.edu.cn

[4] Faculty of Geo-Information and Earth Observation (ITC), University of Twente, 7500 AE Enschede, The Netherlands; l.zhang-2@utwente.nl

\* Correspondence: yaoyunjun@bnu.edu.cn; Tel.: +86-10-5880-3002

**Abstract:** Surface all-wave net radiation (Rn) is a crucial variable driving many terrestrial latent heat (LE) models that estimate global LE. However, the differences between different Rn products and their impact on global LE estimates still remain unclear. In this study, we evaluated two Rn products, Global LAnd Surface Satellite (GLASS) beta version Rn and Modern-Era Retrospective Analysis for Research and Applications-version 2 (MERRA-2) Rn, from 2007–2017 using ground-measured data from 240 globally distributed in-situ radiation measurements provided by FLUXNET projects. The GLASS Rn product had higher accuracy ($R^2$ increased by 0.04–0.26, and RMSE decreased by 2–13.3 W/m$^2$) than the MERRA-2 Rn product for all land cover types on a daily scale, and the two Rn products differed greatly in spatial distribution and variations. We then determined the resulting discrepancies in simulated annual global LE using a simple averaging model by merging five diagnostic LE models: RS-PM model, SW model, PT-JPL model, MS-PT model, and SIM model. The validation results showed that the estimated LE from the GLASS Rn had higher accuracy ($R^2$ increased by 0.04–0.14, and RMSE decreased by 3–8.4 W/m$^2$) than that from the MERRA-2 Rn for different land cover types at daily scale. Importantly, the mean annual global terrestrial LE from GLASS Rn was 2.1% lower than that from the MERRA-2 Rn. Our study showed that large differences in satellite and reanalysis Rn products could lead to substantial uncertainties in estimating global terrestrial LE.

**Keywords:** surface net radiation; terrestrial latent heat flux; GLASS; MERRA-2; uncertainty

## 1. Introduction

Understanding the dynamics of global terrestrial water and carbon fluxes is urgent for the mitigation of climate change, which is characterized by global warming associated with increasing carbon dioxide ($CO_2$) concentrations. Terrestrial latent heat flux (LE), the sum of heat flux from the terrestrial ecosystems to the atmosphere for water evaporation and vegetation transpiration, is a critical component of global energy exchanges [1–3]. Accurate quantification of LE at regional or global scales is essential for water resource management, drought monitoring, and adaptation for climate

change [4–6]. Various terrestrial diagnostic models have been widely used to estimate global and regional LE [7–9].

Surface all-wave net radiation (Rn), defined as the difference between total upward and downward radiation, is a key variable to drive terrestrial diagnostic LE models [1,10]. Rn dominates the energy interchange between the terrestrial ecosystem and the atmosphere, and it has a great influence on the partition of LE and sensible heat flux (H) [10,11]. Although terrestrial LE models can be grouped into statistical models, surface energy balance models—Penman-Monteith models and Priestley–Taylor models [12–17]—require reliable Rn to couple surface conductance for LE estimation; examples include MOD16 [7], GLASS ET [18], GLEAM [19], and ETmonitor [20].

The development of regional and global reliable gridded Rn products is essential for estimating LE over broad scales [21]. There are currently many satellite and reanalysis Rn products available at regional and global scales. Among these Rn products, reanalysis Rn datasets, e.g., MERRA-2, have a high temporal resolution (hourly) but the rather coarse spatial resolution (>20 km) [22]. In contrast, satellite Rn products, e.g., GLASS Rn, have a high spatial resolution (~5 km). These products have been successfully applied to drive ecosystem models and global climate models (GCMs) for ecosystem services and weather/climate forecasting [23,24].

Assessing multiple global Rn products can support us in understanding the accuracies of these products [25]. Several studies have evaluated the accuracy of Rn products using ground-observed data from in situ measurements, and they have found that there are critical uncertainties and differences between several Rn products [26,27]. For example, Jia et al. [28] reported that there were large discrepancies among CERES, EBAF, JRA-55, MERRA-2, and ERA-Interim due to differences in models and input data. Jiang et al. [10] found critical discrepancies between the GLASS Rn product and MERRA-2 Rn in trend, magnitude, and uncertainty. Xin et al. [29] compared CERES and SRB Rn products using Chinese meteorological site data and illustrated that CERES had higher $R^2$ (0.77) versus ground-measurements.

The uncertainty in the global Rn products can lead to uncertainty in terrestrial LE estimation from multiple diagnostic models. Forcing data, model parameterization, and model structure are the primary sources of uncertainty in LE estimation. Previous substantial studies have focused on the uncertainties of estimated LE resulting from input forcing, including satellite parameters (e.g., vegetation index (VI) [30], leaf area index (LAI) [6], and land cover types [31]) and basic meteorological variables (e.g., air temperature (Ta), relative humidity (RH), and vapor pressure deficit (VPD)) [32]. Few studies have evaluated how the uncertainty in Rn products affects the accuracy of LE estimation. Several previous studies have shown that the sensitivity of Rn is highest in many LE models, and the discrepancies in Rn products could lead to considerably different LE estimations. For example, Anderson (2019) reported that satellite Rn products, instead of reanalysis products, improved water flux modeling [33]. However, the impacts of the uncertainty in Rn products on global LE estimation from multiple diagnostic models remain unclear.

In this study, we evaluated Global LAnd Surface Satellite (GLASS) beta version Rn and Modern-Era Retrospective Analysis for Research and Applications, version 2 (MERRA-2) Rn products and detected the impacts of these Rn products on annual global terrestrial LE. We estimated terrestrial LE using a simple model averaging method by merging five diagnostic LE models: remote-sensing-based Penman–Monteith (RS-PM) model, Shuttleworth–Wallace dual-source (SW) model, Priestley–Taylor model of Jet Propulsion Laboratory, Caltech (PT-JPL) model, modified satellite-based Priestley–Taylor (MS-PT) model, and simple hybrid (SIM) model. We had three objectives. First, we validated two GLASS and MERRA-2 Rn products using FLUXNET measurements between 2007 and 2009. Second, we compared the spatial patterns and trends of the two Rn products. Third, we evaluated the effects of Rn on estimated terrestrial LE at the site and global scales.

## 2. Data and Methods

### 2.1. GLASS and MERRA-2 Surface Net Radiation Products

We used two Rn products, including one satellite Rn product (GLASS) and one reanalysis Rn product (MERRA-2). The Global LAnd Surface Satellite (GLASS) beta version Rn was generated using the multivariate adaptive regression splines (MARS) method, moderate-resolution imaging spectroradiometer (MODIS) data, and MERRA-2 meteorological reanalysis data [10]. The MARS model of GLASS Rn was trained by FLUXNET ground-measured Rn data and their corresponding remote sensing data and reanalysis MERRA-2 data [10]. Daily GLASS Rn is a perennial remote sensing product with a spatial resolution of 0.05°, beginning in 2000. This Rn product not only has a better spatial resolution (0.05°) and temporal resolution (daily) but also has higher accuracy than reanalysis data [34]. We, therefore, used the GLASS Rn product as forcing data of the LE model. The GLASS Rn has missing data around the north pole several times a year (most in the spring and winter) because it was produced with remote sensing data. The missing data were temporally filled using the algorithm described by Zhao et al. [35].

The second Rn product was a reanalysis dataset from the Modern-Era Retrospective Analysis for Research and Applications, version 2 (MERRA-2) [22]. The MERRA-2 Rn product was produced by assimilating different observations [34]. MERRA-2 spanned the period from 1980 through 2019 and was produced on a 0.5° × 0.625° grid with an hourly resolution. Though the spatial resolution was coarse, it covered a long-term time span and is spatially and temporally continuous with no missing data. To compare with GLASS Rn, we interpolated MERRA-2 data to 0.05° using a non-linear spatial interpolation method [35]. In addition, daily MERRA-2 data was aggregated from MERRA-2 data with hourly resolution.

### 2.2. Global Terrestrial LE Estimations

#### 2.2.1. LE Models

We used the five diagnostic LE models to simulate LE, and the description of the five diagnostic LE models is demonstrated in Appendix A.

(1)   RS-PM model. The remote sensing-based Penman–Monteith (RS-PM) model was modified from the MODIS global LE model [14]. Mu et al. [7] designed the model by (1) replacing the vegetable cover fraction with a fraction of absorbed photosynthetically active radiation (FPAR), (2) adding night-time LE, (3) estimating soil heat flux, (4) developing estimates of canopy resistance, aerodynamics, and boundary-level, (5) dividing LE into interception evaporation, canopy transpiration, soil evaporation, and wet soil evaporation. Rn, RH, Ta, water pressure (e), and LAI were required to drive the model.

(2)   SW model. The Shuttleworth–Wallace dual-source (SW) model divided LE into soil evaporation and vegetation transpiration. Each component of SW-based LE was calculated by the Penman–Monteith algorithm. The SW model assumed aerodynamic mixing arising at a mean canopy source within the canopy. More detail about the SW model can be viewed elsewhere [36]. The SW model required Rn, RH, Ta, e, wind speed, and LAI.

(3)   PT-JPL model. The Priestley–Taylor of the Jet Propulsion Laboratory (PT-JPL) LE model was proposed by Fisher on the basis of the Priestley–Taylor model [37]. Fisher et al. modified the Priestley–Taylor model using the atmosphere and ecophysiology to calculate the actual LE. The input forcing data to generate PT-JPL LE data was Rn, RH, Ta, e, LAI, and FPAR.

(4)   MS-PT model. The modified satellite-based PT (MS-PT) model was designed by Yao et al. and was based on the PT-JPL model [15]. Yao et al. used the diurnal temperature range (DT) to calculate the apparent thermal inertia (ATI) that represents soil moisture constraints. The MS-PT model divided LE into four components: unsaturated surface soil evaporation, saturated surface

soil evaporation, vegetation canopy transpiration, and vegetation interception evaporation. Since the MS-PT model reduced the parameters of PT-JPL, it only needed Rn, Ta, DT NDVI as inputs.

(5) SIM model. The simple hybrid LE (SIM) model was designed by Wang et al. (2008) by considering the influence of soil moisture on the LE parameterization [38]. This model introduced the influence of soil moisture on the LE parameterization. The coefficients of this model were calibrated using LE measurements in America from 2002 to 2005. The input variables of the SIM model were Rn, Ta, DT, and NDVI.

To examine the effects of Rn on estimated LE, we used the simple model averaging (SA) method to merge five LE models for estimating terrestrial LE. Previous studies have pointed out that the SA method performs better than individual models [18]. The SA method calculated terrestrial LE by averaging each single LE model.

### 2.2.2. Forcing Variables

To simulate LE with two Rn products, we used the MODIS 8-day fraction of photosynthetically active radiation (FPAR) and leaf area index (LAI) product [39] with a spatial resolution of 1 km, and the 8-day average FPAR/LAI was temporally interpolated to daily FPAR/LAI values using linear interpolation. Additionally, the 16-day MODIS normalized difference vegetation index (NDVI) and enhanced vegetation index (EVI) [40] was adopted to drive the LE models. Missing or cloud-contaminated MODIS pixels for NDVI, FPAR, and LAI were temporally filled using the algorithm described by Zhao et al. [35].

Meteorological variables, including vapor pressure (e), relative humidity (RH), diurnal temperature range (DT), and air temperature (Ta), from MERRA-2 data with a spatial resolution of $0.5° \times 0.625°$ were used to drive the LE models. To match GLASS pixels, we used a non-linear spatial interpolation method [35] to interpolate coarse-resolution MERRA-2 data to 0.05-degree GLASS pixels. To evaluate the impact of Rn on LE, we used GLASS Rn and MERRA-2 Rn to drive all LE models.

### 2.3. Comparison and Evaluation of Rn and LE

Rn products and LE models were validated and assessed using ground-measured data from in-situ radiation measurements and eddy covariance flux tower sites. The ground-measured data from 240 in-situ measurements were provided by AsiaFlux, AmeriFlux, LathuileFlux, the Asian Automatic Weather Station Network (ANN) Project, the Chinese Ecosystem Research Network (CERN), and the individual principal investigators (PIs) of the FLUXNET website (https://fluxnet.org/). The in-situ measurements were mostly distributed in North America, Europe, and Asia (Figure 1). We acquired ground-measured data from 240 in-situ measurements, including 34 cropland (CRO) sites, 6 deciduous needleleaf forest (DNF) sites, 28 deciduous broadleaf forest (DBF) sites, 16 evergreen broadleaf forest (EBF) sites, 64 evergreen needleleaf forest (ENF) sites, 10 savanna (SAW) sites, 14 shrubland (SHR) sites, 12 mixed forest (MF) sites, and 56 grass and other types (GRA) sites, between 2001 and 2009. The LE and Rn from ground-measured data can have an error of 10% [41]. The ground-measured datasets included hourly or half-hourly LE, Rn, sensible heat flux (H), and soil heat flux (G) data. The missing data of LE and Rn were gap-filled using the MDS method [42]. Daily LE and Rn values were aggregated from half-hourly and hourly LE and Rn data. Covariance flux towers measured LE by the EC method, which has an issue of energy imbalance [43,44]. To correct the measured LE, we used the method proposed by Twine et al. [42]. The method can be written as

$$LE_C = \frac{LE}{ECR}, \tag{1}$$

$$ECR = \frac{LE + H}{Rn - G}, \tag{2}$$

where $LE_C$ is the corrected LE. ECR is the ratio of energy closure, and LE is the uncorrected LE.

$R^2$, RMSE, and Bias were adopted to evaluate the accurateness of individual LE models and the SA method. $R^2$ can be calculated as the square of the correlation coefficient between estimations and observations. Bias is the mean value of the discrepancy between estimations and observations. It can be written as

$$\text{Bias} = \frac{1}{M} \sum_{i=1}^{M} (E_i - T_i), \tag{3}$$

where $M$ is the number of samples, $E_i$ is the value of estimations, and $T_i$ is the value of observations. RMSE is the standard error between estimations and observations. It can be expressed as

$$\text{RMSE} = \sqrt{\frac{1}{M} \sum_{i=1}^{M} (E_i - T_i)^2}, \tag{4}$$

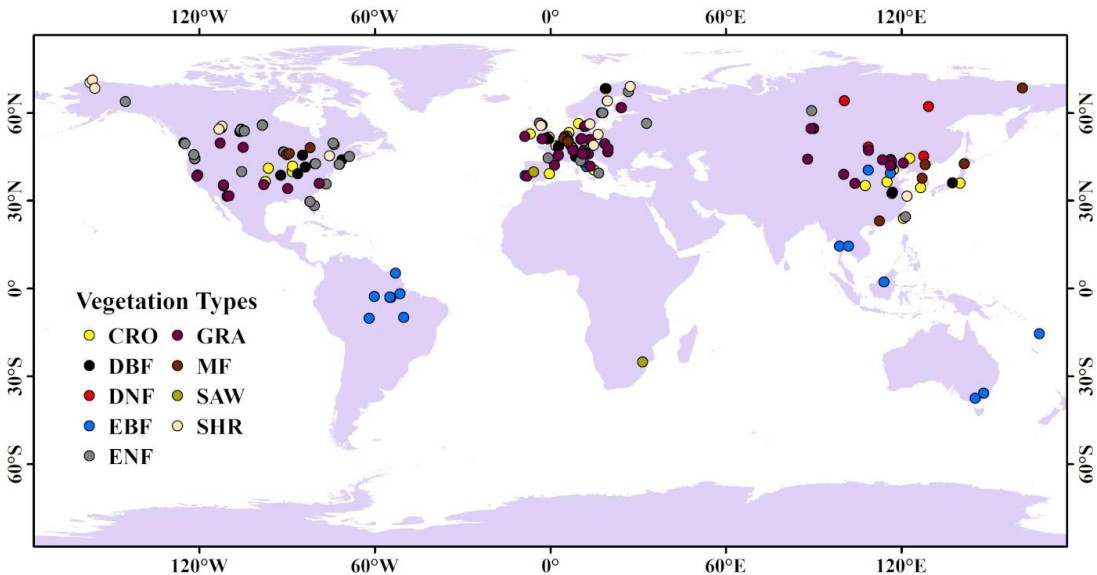

**Figure 1.** Distribution of the in-situ measurements used in this study.

## 3. Results

*3.1. Validation of GLASS and MERRA-2 Rn Products with Ground-Measured Data*

To evaluate the quality of the two Rn products, we used the ground-measured data from 240 in-situ radiation measurements with different land cover types. At the site scale, we found there were large discrepancies in the two Rn products among different land cover types (Figure 2). For both Rn products, ENF sites had the highest $R^2$ (0.83 for GLASS and 0.74 for MERRA-2) compared with other land cover types, whereas the lowest RMSEs were SHR for GLASS (27.5 W/m$^2$) and GRA for MERRA-2 (38.7 W/m$^2$). The two Rn products exhibited lower $R^2$ (0.57–0.70 for GLASS and 0.39–0.57 for MERRA-2) with higher RMSE (35.5–48.0 W/m$^2$ for GLASS and 46.2–50.9 W/m$^2$ for MERRA-2) in DNF, EBF, and SAW sites. In addition, the GLASS Rn was underestimated when the ground-measured value was high. This phenomenon might result from the method the GLASS Rn product adopted. Since GLASS Rn used MARS to train their model, the performance of machine learning was highly related to data [10].

Overall, the GLASS Rn accounted for 57–83% of Rn variability for all ground-measured data, whereas the MERRA-2 Rn only explained 39–69% of Rn variability (Figure 2). The GLASS Rn product had better performance than the MERRA-2 Rn product ($R^2$ increased by approximately 0.10, and RMSE decreased by approximately 7.7 W/m$^2$). The higher accuracy of the GLASS Rn might result from the finer spatial resolution of the GLASS product.

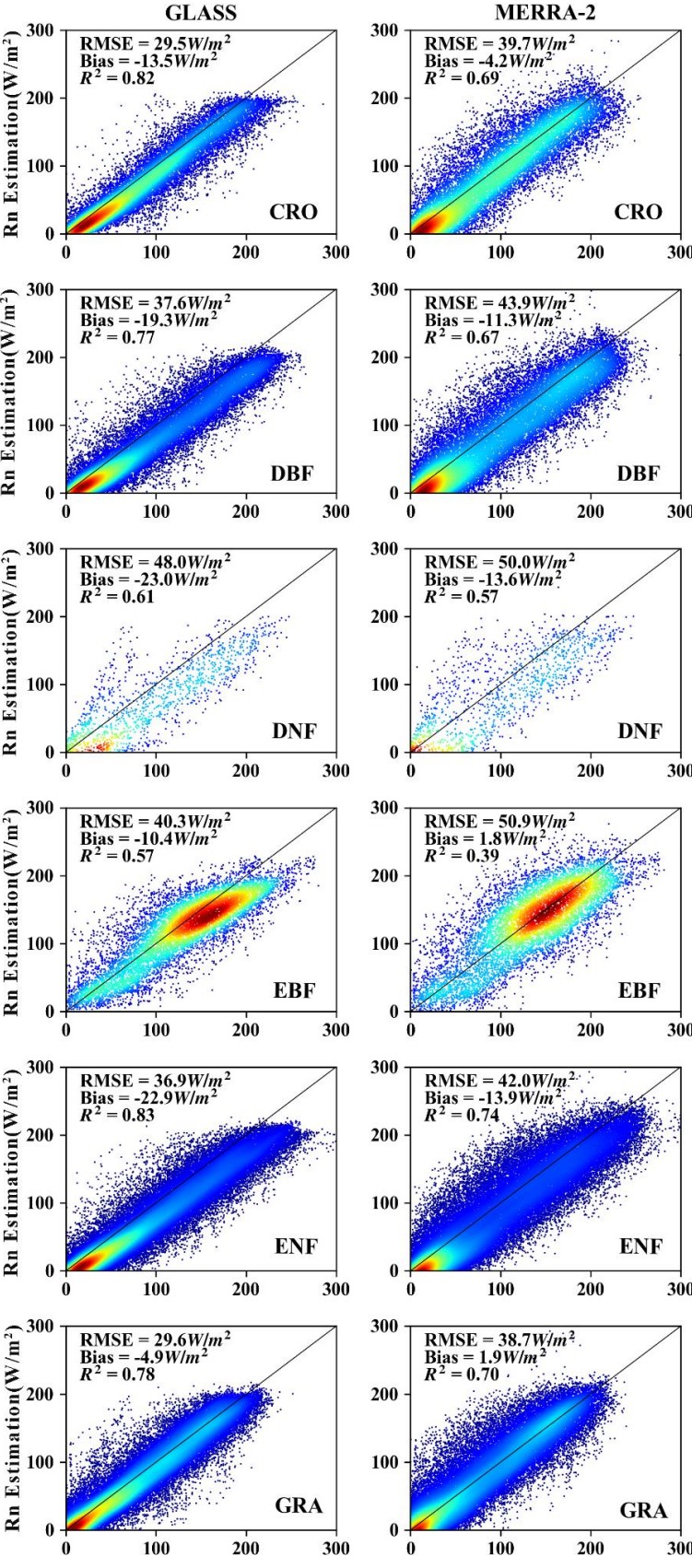

**Figure 2.** *Cont.*

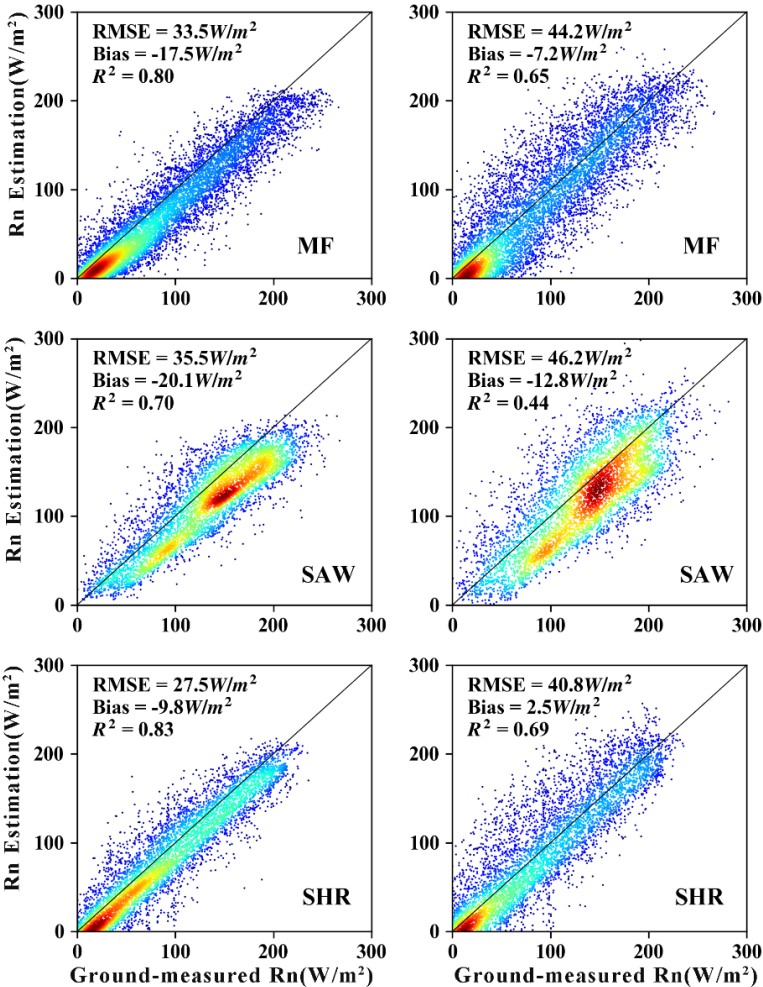

**Figure 2.** Rn scatter plots between ground-measured Rn (*x*-axis, unit: W/m²) and the estimated Rn from two products (GLASS and MERRA-2) (*y*-axis, unit: W/m²) at 240 in-situ radiation measurements for different land cover types. Rn, radiation; GLASS, Global LAnd Surface Satellite; MERRA-2, Modern-Era Retrospective Analysis for Research and Applications-version 2.

### 3.2. Spatial Differences of GLASS and MERRA-2 Rn Products

Figure 3 demonstrates the spatial pattern of annual mean Rn and discrepancy for GLASS Rn and MERRA-2 Rn over the period from 2007–2017. The two Rn products were characterized by similar spatial Rn distributions. For example, the Central Africa, South America, and Southeast Asia regions showed the highest Rn, whereas North Africa and high latitude areas exhibited the lowest Rn owing to solar radiation limitations and their high surface albedo. A difference in Rn spatial patterns was also observed. For example, the Andes region had high Rn values for MERRA-2 and medium values for GLASS; the Tibetan Plateau exhibited medium values for MERRA-2 and low values for GLASS; the Sahara had higher values for MERRA-2 than GLASS. The magnitude of Rn varied between the two Rn products. Most of the regions showed negative differences in Rn between GLASS Rn and MERRA-2 Rn, indicating that MERRA-2 Rn had higher values.

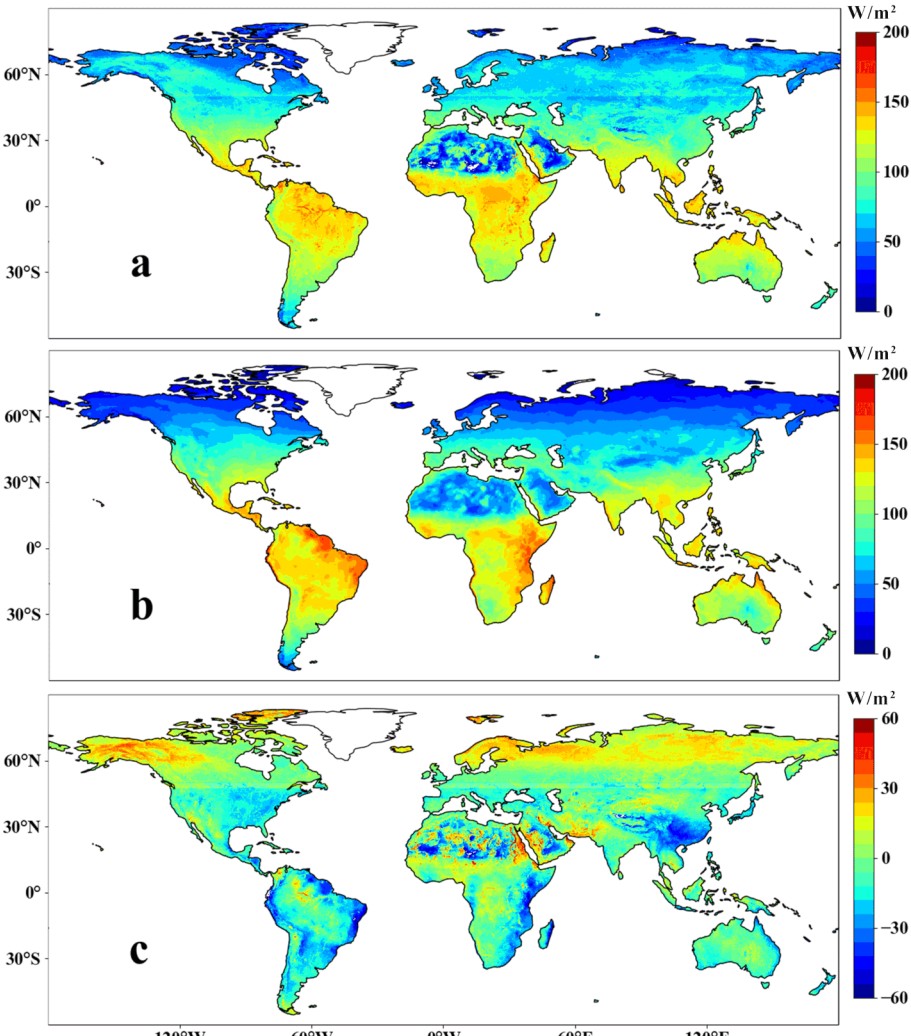

**Figure 3.** Spatial patterns of average annual Rn (W/m$^2$) of GLASS Rn (**a**) and MERRA-2 Rn (**b**) in the period from 2007 to 2017 and their difference (GLASS-MERRA-2, (**c**)).

Figure 4 shows the Rn trend of the two Rn products during 2007–2017. Rn decreased over 4.5% and 18.2% of global land area for GLASS and MERRA-2, respectively, and increased over 3.8% and 10.6% of global land area for MERRA-2 and GLASS. Generally, GLASS Rn exhibited an increasing trend in most regions of the global land area, and MERRA-2 Rn showed a decreasing trend. The decreasing MERRA-2 Rn might indicate sparse vegetation, and severe desertification occurred in Central Africa. The different spatial patterns in Rn trends of the two Rn products might be caused by different model structure and input data of two Rn products [10,34]. Figure 5a presents the trends of globally averaged annual Rn of the two Rn products during the period 2007–2017. The globally averaged annual GLASS Rn exhibited insignificant upward trends. In contrast, MERRA-2 Rn showed insignificant decreasing trends. The two globally averaged annual Rn had small discrepancies that ranged from 0.3 W/m$^2$ to 1.6 W/m$^2$. Figure 5b shows that the zonal means of two Rn products during the period from 2007–2017 changed with the latitude. Despite the similar spatial distributions in the two Rn products, MERRA-2 yielded higher Rn between latitudes of 55°S and 47°N, whereas GLASS had higher Rn in high latitude areas.

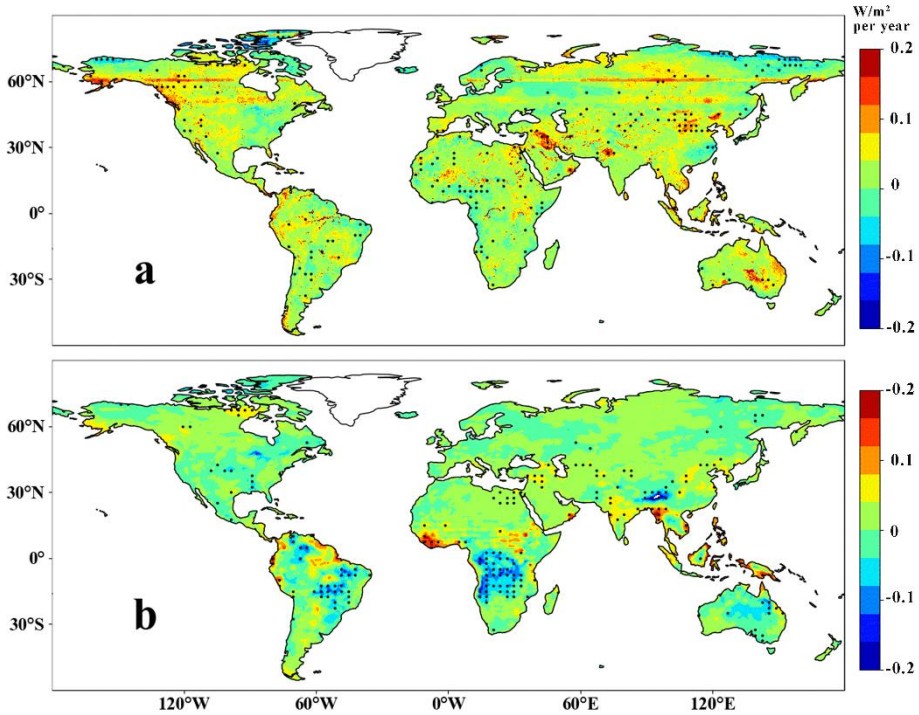

**Figure 4.** Trends of GLASS Rn (**a**) and MERRA-2 Rn (**b**) during the period of 2007–2017. The black points refer to pixels with 95% confidence.

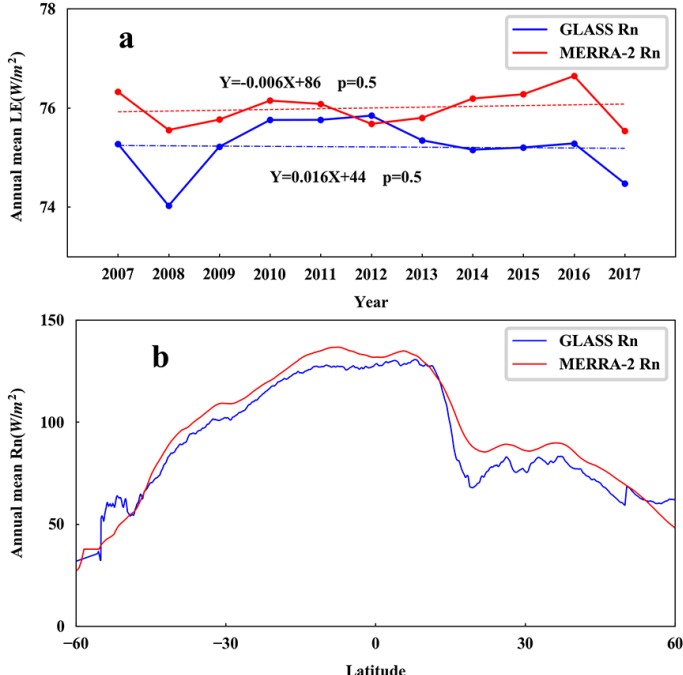

**Figure 5.** Trends in globally-averaged annual Rn (**a**) and latitude series of average Rn (**b**) for two Rn products during the period 2007–2017.

### 3.3. Validation of Simulated LE Driven by GLASS and MERRA-2 Rn Products with EC Observations

LE data simulated by six models were also validated using flux tower sites (Table 1). The GLASS-based LE had higher $R^2$ and lower RMSE compared to the MERRA-2-based LE for different LE models ($R^2$ increased by 0.04–0.11, and RMSE decreased by 3–6.8 W/m$^2$). For GLASS-based LE, both the PT-JPL and SA models had the highest $R^2$ (0.6) and better performance, whereas SA

exhibited lower RMSE than the PT-JPL model. The SW model had the lowest $R^2$ (0.53) and the worst RMSE (31.7 W/m$^2$) for GLASS-based LE. For MERRA-2-based LE, the SA model also had the highest $R^2$ (0.52) and the lowest RMSE (31.8 W/m$^2$). The SIM model exhibited the lowest $R^2$ (0.48) for MERRA-2-based LE, yet the SW model showed the highest RMSE. We found that the SA model had better performance than individual terrestrial LE models for both GLASS-based LE and MERRA-2-based LE. We, therefore, validated the LE of the SA model using flux tower sites for different land cover types.

**Table 1.** Table of statistics (RMSE and $R^2$) of the comparison between LE estimation from multiple LE models (based on MERRA-2 Rn and GLASS Rn) and ground-measured at 240 flux tower sites.

| Rn Products | Models | $R^2$ | RMSE (W/m$^2$) |
|---|---|---|---|
| GLASS | RS-PM | 0.56 | 28.8 |
| | SW | 0.53 | 31.7 |
| | PT-JPL | 0.6 | 27.4 |
| | MS-PT | 0.59 | 27.8 |
| | SIM | 0.59 | 27.2 |
| | SA | 0.6 | 26.6 |
| MERRA-2 | RS-PM | 0.5 | 33.1 |
| | SW | 0.49 | 34.7 |
| | PT-JPL | 0.49 | 34.2 |
| | MS-PT | 0.49 | 33.1 |
| | SIM | 0.48 | 33.2 |
| | SA | 0.52 | 31.8 |

Figure 6 presents the scatter plot between the LE estimations from the two Rn products using the SA model and ground-measured LE for different land cover types. At the site scale, we found a large difference in the LE estimation from the two Rn products among different land cover types. For the EBF sites, both GLASS-based LE and MERRA-2-based LE had the lowest $R^2$ (0.47 and 0.35) and the highest RMSE (32 W/m$^2$ and 40.4 W/m$^2$). For the CRO sites, GLASS-based LE had better $R^2$ (0.55) and lower RMSE (31 W/m$^2$) than MERRA-2-based LE ($R^2$: 0.44, RMSE: 35.1 W/m$^2$). GLASS-based LE had the highest $R^2$ (0.72) for SAW sites, whereas MERRA-2-based LE exhibited the highest $R^2$ (0.64) for MF sites. For the DNF sites, both GLASS-based LE and MERRA-2-based LE presented the lowest RMSE (20.9 W/m$^2$ and 23.9 W/m$^2$). For different land cover types, the comparisons between the GLASS-based LE and MERRA-2-based LE showed that the GLASS-based LE had better performance. The values of $R^2$ for LE estimated by GLASS (0.47–0.72) were much higher than LE simulated by MERRA-2 (0.35–0.64). Similarly, the values of RMSE for LE estimated by GLASS (20.9–31 W/m$^2$) were much lower than LE simulated by MERRA-2 (23.0–35.1 W/m$^2$). In comparison to ground-measured LE data, most land types of the two LE products (except CRO) yielded higher values than ground-measured LE. For the CRO sites, the two products underestimated the values of LE since they did not consider the influence of irrigation on LE. Although the GLASS-based LE did not result in a better bias for CRO, it still had higher $R^2$ and lower RMSE than MERRA-2-based LE. For all land cover types, GLASS-based LE had better accuracy than MERRA-2-based LE. This indicated that the performance of the Rn product affected the accuracy of the LE product.



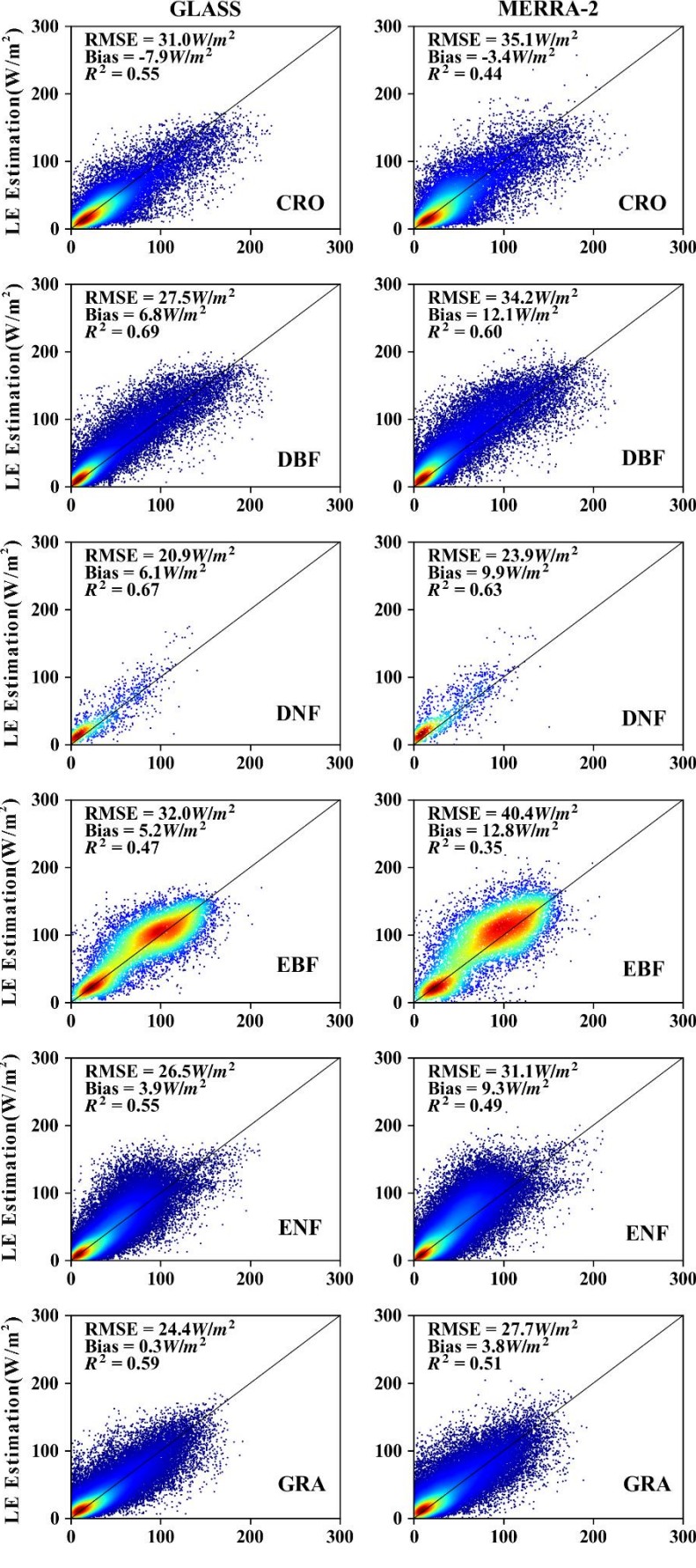

**Figure 6.** *Cont.*

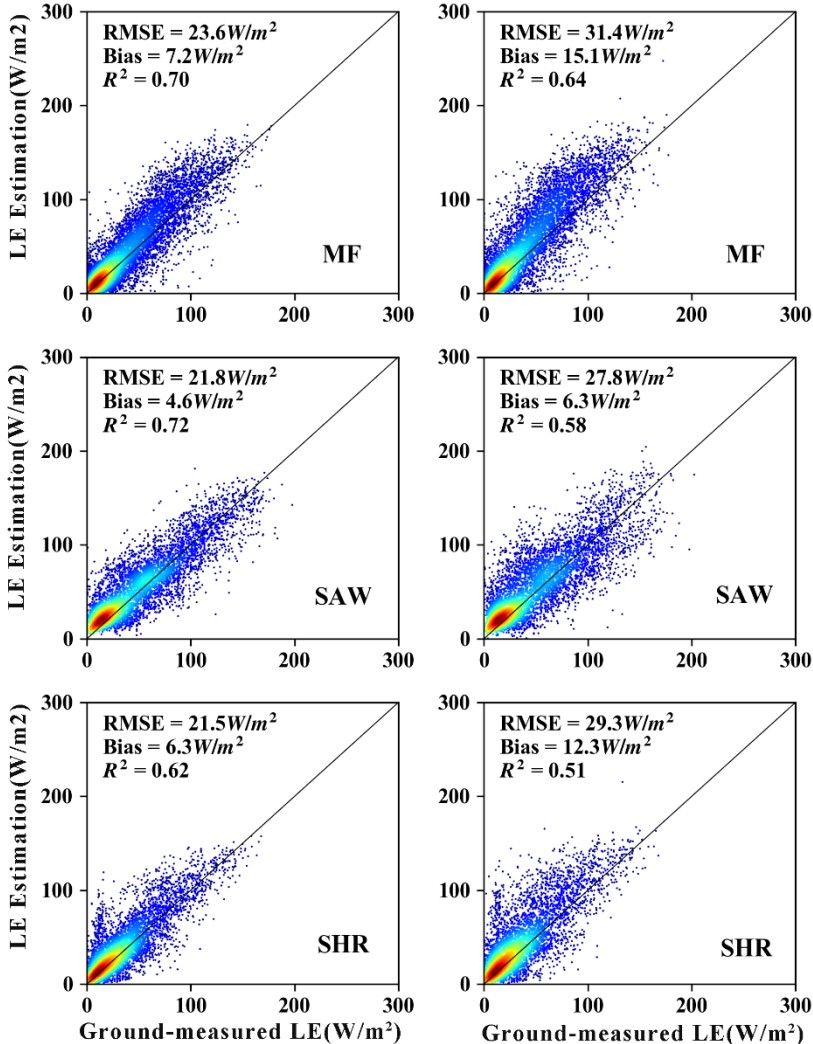

**Figure 6.** Scatter plots between ground-measured latent heat (LE) (*x*-axis, unit: W/m$^2$) and the estimated LE using the simple averaging (SA) method (*y*-axis, unit: W/m$^2$) at 240 flux tower sites for different land cover types.

## *3.4. Spatial Comparisons of Simulated LE Based on GLASS and MERRA-2 Rn Products*

The estimated average annual LE from 2007 to 2017 based on the two Rn products was generally characterized by similar spatial patterns (Figure 7): LE had higher values in Central Africa, South America, and Southeast Asia regions. The patterns were mainly resolved by the spatial distribution of land surface characteristics and climate factors. High-temperature and sufficient water in these regions could provide suitable conditions for evapotranspiration. However, arid and desert regions had the lowest LE due to water limitations and sparse vegetation. For example, both the GLASS-based LE and the MERRA-2-based LE showed low values in the Sahara. Despite similar spatial patterns, the average annual LE based on the two Rn products showed obvious discrepancies in some regions (Figure 8). For example, the Andes region had high LE values for MERRA-2 and middle-level values for GLASS; the south of China exhibited high values for MERRA-2 and middle-level values for GLASS.

Figure 8 shows the trends of estimated annual LE in the period from 2007 to 2017. Annual MERRA-2-based LE increased in Central Africa with increasing MERRA-2 Rn, whereas annual GLASS-based LE decreased with increasing GLASS Rn. The difference in trends in estimated annual LE resulted from discrepancies in the trends of different Rn products. Both GLASS-based LE and MERRA-2-based LE showed significant increases in southeast Asia and significant decreases in

north-western Australia. GLASS-based LE and MERRA-2-based LE increased over 21.3% and 12.4% of the global land area and decreased over 4.6% and 7.5% of the global land area, respectively.

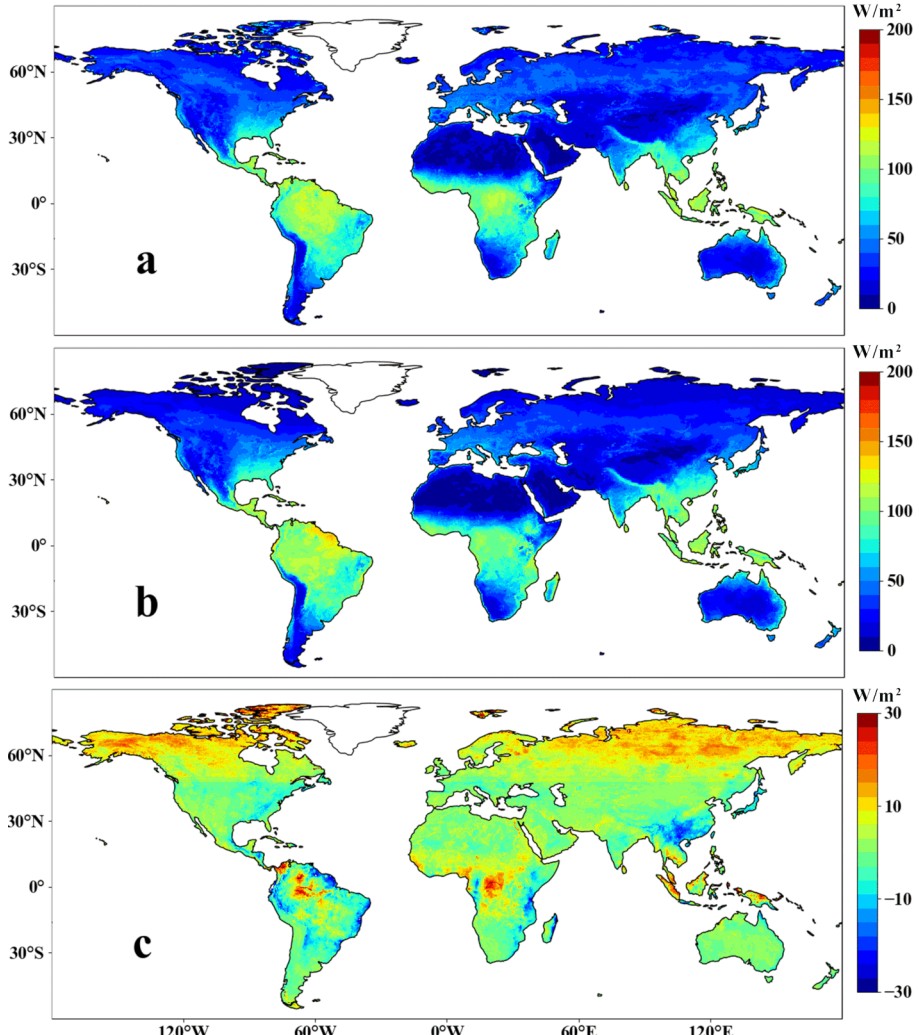

**Figure 7.** Spatial patterns of average annual LE (W/m$^2$) of GLASS-based LE (**a**) and MERRA-2-based LE (**b**) in the period from 2007 to 2017 and their difference (GLASS- MERRA-2, (**c**)).

Figure 9a illustrates the trends of globally averaged annual LE based on the two Rn products during the period 2007–2017. The globally averaged LE showed significant upward trends for the two Rn products. The two globally averaged annual LE exhibited small discrepancies that ranged from 0.4 W/m$^2$ to 1.39 W/m$^2$. The GLASS globally averaged annual LE ranged from 35.8 to 37.7 W/m$^2$ compared with MERRA-2 globally averaged annual LE from 36.2 to 38.2 W/m$^2$. The GLASS globally averaged annual LE was slightly lower than MERRA-2 globally averaged annual LE. Figure 9b shows that the zonal means of two LE products during the period from 2007–2017 changed with latitude. MERRA-2-based LE had higher values between latitudes of 55°S and 47°N, whereas GLASS-based LE had higher values in high latitude areas.

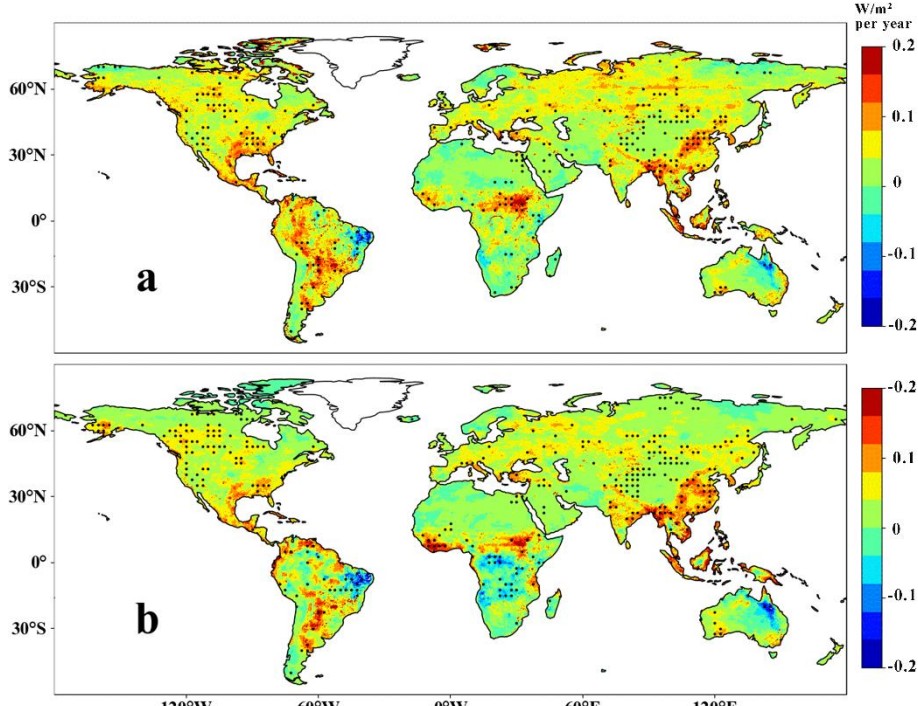

**Figure 8.** Trends of annual LE (W/m$^2$) of GLASS-based LE (**a**) and MERRA-2-based LE (**b**) during the period of 2007–2017. The black points refer to pixels with 95% confidence.

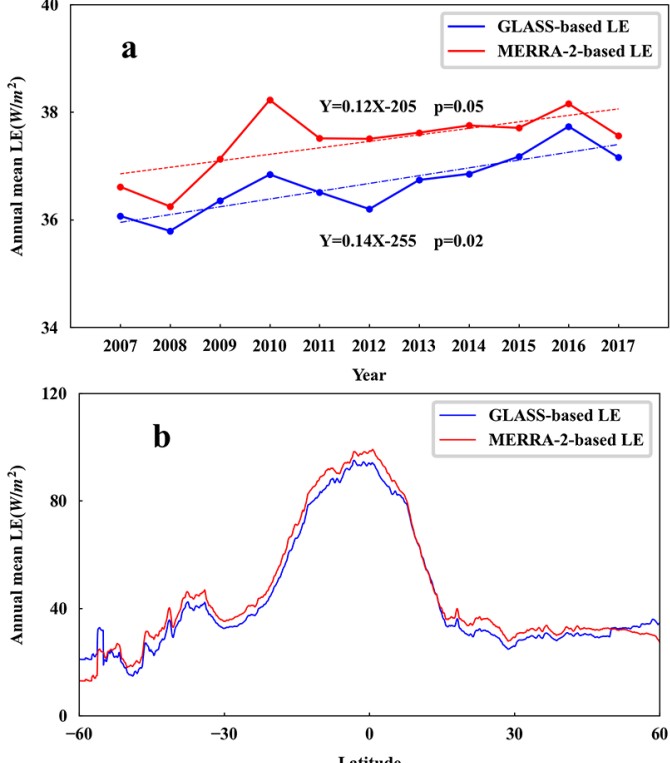

**Figure 9.** Trends in globally-averaged annual LE (**a**) and latitude series of average LE (**b**) based on the two Rn products during the period 2007–2017.

## 4. Discussion

We evaluated the accuracy of two Rn products on a global scale during the period from 2007–2017. Many studies have compared multiple Rn products. For example, Jiang et al. found a large gap between CERES-SYN, MERRA-2, JRA55, and GLASS Rn products that were validated using in situ observation data [10]. Jia et al. found great discrepancies in the magnitude and spatial pattern of Rn among CERES, ERA-Interim, MERRA-2, and JRA-55 products from 2001 to 2013 [28]. Our study showed that GLASS and MERRA-2 Rn products exhibited great discrepancies in magnitude and spatial patterns. This research indicated that different Rn products had differences in spatial patterns and magnitude. We found that the discrepancy of Rn products could lead to differences in long-term trends in Rn. Many previous researchers have compared and validated different Rn products, but these comparisons and validations have been simply done using whole data sets [45–47]. In this study, we assessed two Rn products using covariance flux tower data that was separated into several land cover types. The accuracy of Rn products varied for the diversity of land cover types because the differences in surface biophysical parameters (e.g., surface albedo, surface emissivity, soil moisture) across multiple biomes alter surface energy budget. In general, the errors of GLASS Rn were lower than MERRA-2 Rn for each land cover type.

The two Rn products have three major sources of uncertainty: input data, the algorithm of products, and spatial scale mismatch [48,49]. The retrieval method and input data could cause large discrepancies among the two Rn products. Since the two Rn products use completely different algorithms and data, they have a substantial difference in magnitude and spatial patterns [10,22]. The method of GLASS Rn employs MARS to train the Rn model using reanalysis MERRA-2 data, remote sensing data, and ground-measured data [10]. The accuracy of the GLASS Rn model is mainly dependent on the training data. Thus, the model of GLASS Rn is determined by sample distribution and sample size [50]. The input data of GLASS Rn, such as Ta, surface albedo, surface air pressure, and wind speed, are also key sources of uncertainty. The error of those data could cause errors and unreasonable values for Rn [26]. The MERRA-2 uses an underlying forecast model to fuse disparate observations, enabling the production of gridded datasets, such as Rn, RH, Ta, and so on. The input observations of MERRA-2 include conventional observations, satellite observations of wind, satellite retrievals of temperature, satellite retrievals of rain rate, and ozone retrievals [22]. The input observations of MERRA-2 are important to the uncertainty of MERRA-2, and the quality of the MERRA-2 data is related to the quality of the input observations. The pixel average for MERRA-2 Rn is $0.5° \times 0.625°$, whereas eddy covariance flux tower sites can represent a tiny scale of several hundred meters, which cause large differences between Rn observations and the MERRA-2 Rn [23,51,52]. However, the spatial resolution of GLASS Rn ($0.05° \times 0.05°$) is finer than the MERRA-2 Rn and is closer to the footprint of eddy covariance flux tower sites. That may lead to higher accuracy of the GLASS Rn [53]. The topography (e.g., aspect, slope, elevation,) is also a critical factor for Rn simulation. The accurately topographical information can help us estimate reasonable direct solar radiation [54]. Both the GLASS and MERRA-2 Rn products do not consider the influence of topography on Rn. We will use topographical information to improve Rn estimation in the future.

The comparison and assessment of the two Rn products can make us explore how the use of the two products influences estimated LE. Previous research adopted site-scale sensitivity analyses and found that simulated LE is sensitive to Rn [25]. However, our research inspected the effects of gridded Rn on estimated LE on a global scale. Our research found that the differences in the two Rn products led to obvious discrepancies in the magnitude of LE at the global scale. These discrepancies in Rn could lead to larger discrepancies in the estimation of LE. The accurate quantification of LE has significant implications for managing water resources [15,55]. The differences in Rn among the two Rn products also lead to discrepancies in the trends and spatial patterns of LE estimated by the SA method. An accurate Rn is important for realistically estimating the trends and spatial patterns of LE and better managing water resources.

The estimation of the LE model has three origin sources of uncertainty: input data of model, model structure, and model parameters [56–59]. To retrieve LE, we used a simple model averaging method replacing individual LE algorithms since the results illustrated that the SA model had better performance than individual LE models. For different land cover types, the discrepancies in LE estimation varied greatly due to the difference in surface moisture status and vegetation stomatal conductance. Our results indicated that the Rn data could also lead to critical uncertainties in estimated LE, demonstrating that one of the significant sources of uncertainty of LE models is Rn [51]. We need to find the relative effects of Rn, vegetation index, and meteorological data on the LE model in further research. The GLASS Rn had higher accuracy and led to more accurate LE, indicating that the accuracy of Rn could influence the estimation of LE. Our work suggested that the GLASS Rn might be more suitable for use with LE models when estimating LE on a global scale.

Rn was used in our SA model, and the difference in the Rn products led to important uncertainty in the result of LE estimates. As our results demonstrated, the GLASS Rn had fewer uncertainties than the MERRA-2 Rn and could yield more accurate and reasonable LE on a global scale. The development of global, accurate, and long-term Rn products will increase the accuracy of global LE estimation.

## 5. Conclusions

In this study, we first assessed the accuracy of two Rn products using 240 ground-measured data. Our result showed that GLASS Rn had higher accuracy than MERRA-2 Rn for all land cover types on a daily scale ($R^2$ increased by 0.04–0.26, and RMSE decreased by 2–13.3 W/m$^2$). The daily Rn from GLASS product agreed better with the ground-measured data for different land cover types, indicating GLASS Rn had better performance than MERRA-2 Rn. The differences between the GLASS Rn and MERRA-2 Rn led to a large discrepancy in the estimation of annual LE on a global scale. The mean annual LE for the whole globe based on the GLASS Rn was 2% lower than that based on the MERRA-2 Rn. During the period from 2007–2017, the annual LE simulated from the GLASS Rn increased by over 3%, while it only increased by 2.5% from the MERRA-2 Rn.

Overall, we conclude that we can use GLASS Rn to improve the accuracy of daily LE estimation compared with MERRA-2 Rn. The high accuracy of Rn products can obtain a more reliable and reasonable estimation of terrestrial LE over the globe.

**Author Contributions:** Conceptualization, Y.Y.; methodology, Y.Z.; validation, Y.L. and B.J.; formal analysis, K.J.; investigation, X.Z.; resources, X.X. and L.Z.; data curation, K.S., J.Y., and X.B.; writing—original draft preparation, X.G. All authors have read and agreed to the published version of the manuscript.

**Funding:** This work was partially supported by the National Key Research Development Program of China (No.2016YFA0600103 and No. 2016YFB0501404) and the Natural Science Fund of China (No.41671331 and No. 41701483).

**Acknowledgments:** We would like to thank Shaomin Liu and Ziwei Xu from Beijing Normal University, China, and Guangsheng Zhou from the Institute of Botany, CAS, and Yan Li and Ran Liu from Xinjiang Institute of Ecology and Geography, CAS, and Guoyi Zhou and Yuelin Li from South China Botanic Garden, CAS, and Bin Zhao from Fudan University, China, for providing ground-measured data. This work used eddy covariance data acquired by the FLUXNET community and, in particular, by the following networks: AmeriFlux (U.S. Department of Energy, Biological and Environmental Research, Terrestrial Carbon Program (DE-FG02-04ER63917 and DE-FG02-04ER63911)), AfriFlux, AsiaFlux, CarboAfrica, CarboEuropeIP, CarboItaly, CarboMont, ChinaFlux, FluxnetCanada (supported by CFCAS, NSERC, BIOCAP, Environment Canada, and NRCan), GreenGrass, KoFlux, LBA, NECC, OzFlux, TCOS-Siberia, USCCC. We acknowledge the financial support to the eddy covariance data harmonization provided by CarboEuropeIP, FAO-GTOS-TCO, iLEAPS, Max Planck Institute for Biogeochemistry, National Science Foundation, University of Tuscia, Université Laval, Environment Canada and US Department of Energy and the database development and technical support from Berkeley Water Center, Lawrence Berkeley National Laboratory, Microsoft Research eScience, Oak Ridge National Laboratory, University of California-Berkeley, and the University of Virginia. Other ground-measured data were obtained from the GAME AAN (http://aan.suiri.tsukuba.ac.jp/), the Coordinated Enhanced Observation Project (CEOP) in arid and semi-arid regions of northern China (http://observation.tea.ac.cn/), and the water experiments of Environmental and Ecological Science Data Center for West China (http://westdc.westgis.ac.cn/water). MODIS LAI/FPAR, NDVI, Albedo, and land cover satellite products were obtained online (http://reverb.echo.nasa.gov/reverb).

**Conflicts of Interest:** The authors declare no conflict of interest.

## Appendix A

Mu et al. [7] developed the remote sensing-based Penman–Monteith model by modifying the Penman–Monteith logic. The RS-PM can be calculated as follows:

$$LE = \frac{\Delta(R_n - G) + \rho C_p VPD / r_a}{\Delta + \gamma(1 + r_s / r_a)}, \tag{A1}$$

where $\Delta$ is the slope of the relationship of saturation vapor pressure and air temperature, $\rho$ is the air density, $\gamma$ is the psychrometric constant, $C_p$ is the specific heat of the air, $VPD$ is the vapor pressure deficit, $r_s$ is the surface resistances, and $r_a$ is the aerodynamic resistance.

The SW model divides LE into soil evaporation and vegetation transpiration [36], and the model can be written as:

$$LE = LE_s + LE_v, \tag{A2}$$

$$LE_s = \frac{\Delta(R_n - G) + \left(\rho C_p VPD - \Delta r_{as} R_{nc}\right)/(r_{aa} + r_{as})}{\{\Delta + \gamma[1 + r_{ss}/(r_{aa} + r_{as})]\}\{1 + [R_s R_a/(R_c(R_s + R_a))]\}}, \tag{A3}$$

$$LE_v = \frac{\Delta(R_n - G) + \left[\rho C_p VPD - \Delta r_{ac}(R_{ns} - G)\right]/(r_{aa} + r_{ac})}{\{\Delta + \gamma[1 + r_{sc}/(r_{aa} + r_{ac})]\}\{1 + [R_c R_a/(R_s(R_c + R_a))]\}}, \tag{A4}$$

$$R_a = (\Delta + \gamma)r_{aa}, \tag{A5}$$

$$R_s = (\Delta + \gamma)r_{as} + r_{ss}\gamma, \tag{A6}$$

$$R_c = (\Delta + \gamma)r_{ac} + r_{sc}\gamma, \tag{A7}$$

where $LE_s$ and $LE_a$ are the soil evaporation and vegetation transpiration, $r_{aa}$ is the aerodynamic resistance from reference highness to vegetation highness, $r_{ac}$ is the aerodynamic resistance from foliage to canopy highness, $r_{as}$ is the aerodynamic resistance from the surface to canopy highness, $r_{ss}$ is the surface resistance of soil, and $r_{sc}$ is the surface resistance of vegetation.

The PT-JPL LE model was proposed by Fisher on the basis of the Priestley–Taylor logic [37], and it can be calculated as follows:

$$LE = LE_s + LE_c + LE_i, \tag{A8}$$

$$LE_s = 1.26[f_{wet} + (1 - f_{wet})f_{sm}]\frac{\Delta}{\Delta + \gamma}(R_{ns} - G), \tag{A9}$$

$$LE_c = \alpha(1 - f_{wet})f_g f_T f_M \frac{\Delta}{\Delta + \gamma}R_{nc}, \tag{A10}$$

$$LE_i = \alpha f_{wet}\frac{\Delta}{\Delta + \gamma}R_{nc}, \tag{A11}$$

where $LE_S$, $LE_c$, and $LE_i$ are soil evaporation, vegetation transpiration, and evaporation of canopy interception. $f_{wet}, f_{sm}$, and $f_g$ are the surface wet fraction ($RH^4$), moisture constraint of soil ($RH^{VPD}$), and canopy fraction ($f_{APAR}/f_{IPAR}$). $f_T$ and $f_M$ are the constraints of temperature and constraint of moisture. Rns and Rnc are Rn of soil and Rn of vegetation.

The MS-PT model was developed by Yao et al. [15], and it can be written as:

$$LE = LE_{ds} + LE_{ws} + LE_v + LE_{ic}, \tag{A12}$$

$$LE_{ds} = 1.26[f_{wet} + (1 - f_{wet})f_{sm}]\frac{\Delta}{\Delta + \gamma}(R_{ns} - G), \tag{A13}$$

$$LE_v = 1.26(1 - f_{wet})f_c f_T \frac{\Delta}{\Delta + \gamma}R_{nc}, \tag{A14}$$

$$LE_{ic} = 1.26 f_{wet} \frac{\Delta}{\Delta + \gamma} R_{nc}, \tag{A15}$$

$$f_{sm} = \left(\frac{1}{DT}\right)^{DT/40}, \tag{A16}$$

$$f_{wet} = f_{sm}^4, \tag{A17}$$

$$f_c = \frac{NDVI - NDVI_{min}}{NDVI_{max} - NDVI_{min}}, \tag{A18}$$

where $LE_{ds}$, $LE_{ws}$, $LE_v$, and $LE_{ic}$ are the soil evaporation, wet soil evaporation, vegetation transpiration, and interception evaporation. $NDVI_{min}$ and $NDVI_{max}$ are 0.05 and 0.95 [15].

Wang et al. developed the SIM model [38], and it can be calculated as follows:

$$LE = R_n(0.144 + 0.6495 * NDVI + 0.009 * Ta + 0.0163 * DT), \tag{A19}$$

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
