# Peer review of "Discrepancies in the Simulated Global Terrestrial Latent Heat Flux from GLASS and MERRA-2 Surface Net Radiation Products"

_remotesensing, doi:10.3390/rs12172763_

Round 1
Reviewer 1 Report
The authors present a detailed evaluation surface all-wave net radiation measurements from 240 sites with a good geographical and climatic region spread. They compare the measured data to the projections one would obtain from two major products commonly utilised for this purpose (GLASS and MERRA-2), and apply a variety of models to each of these.
Their results are credible and their findings useful. The work therefore deserves publication. The paper is also extremely well written, and does not require major corrections.
There are two points that I feel the authors should explain in more detail, alternatively develop further:
i) The origin, nature and selection of the data is extremely unclear. While there is a map illustrating the locations used for the measurements, it is currently not possible to identify the sites with confidence. I am not suggesting a table with 240 entries, but rather some sort of a web link to enable an interested reader to access the list of sites. Or at least there needs to be an explanation about the criteria used to select the sites. If the study includes all sites for which such data exists (unlikely), then the paper should state that. Related to that, and perhaps even more important, there need to be details about how much data was used from each site, and how that data was selected (again, if every available data point between 2007 and 2017 was used, then the paer must state that). It says somewhere that the data used are daily averages. Is it possible that there is oversampling at particular times of the year, and undersampling at others? If one site has mainly winter data and another one mainly summer data, then that intruduces a bias that may influence some of the regional or latitude dependences observed. I am not suggesting a repitition of the analysis on a seasonal basis, but rather to consider this effect in the evaluation of the results.
ii) Related to the last sentence above, I think that the authors should go further in exploring the trends they observed in Section 3.2. Instead of just stating the regional peculiarities, offer some explanations for why these might be manifesting themselves. I don't feel that what is written in the paper at the moment goes far enough. For example, is it possible that the regional anomalities and trends over time are in any way related to drought/wet periods?
Then I just have a list of minor corrections or suggestions:
Line 26: I think "determined" is more accurate here than "detected".
Lines 28-30: I think leave out the acronyms "SA", "RS-PM", "PT-JPL", etc.. Abstracts should be used for an overview, and not to define acronyms.
Line 42: "CO2" should not be in italics.
Line 80: I guess you mean "Anderson et al.", and not "Martha".
Line 101, 102 & 109: use a proper degree sign (as in line 107), not "o".
Equation 1 (and also at many other places in the paper): Please be consistent throughout when using italics (and when not using them). If "LE" (or any other of the short forms used in this paper, e.g. :Ta") is a variable parameter, then it would normally be in italics throughout the paper. If it is just an abbreviation, then it should not be in italics.
Line 129: replace "acquired from this research" to "viewed elsewhere".
Figs. 2 and 7: I personally think that these presently take up too much space, and I'm not sure if these plots are all necessary. To me they tell little more than that some climatic regions are far better sampled than others.
Lines 252-264: Don't you think that this would look much better in the form of a table?
Figure 10: The horizontal ordinates -55, -25, 5, ... are rather odd. I think -60, -30, 0, 30, 60 would look far better (and is also more natural).
Line 413 and 505: I guess you mean "IEEE". Also check that the journal abbreviation used are the standard ones used by "Remote Sensing" journal.
Line 499: Clearly author Gao does not have the initials "HJIJoRS". The journal name is also missing.
Line 513: The characters given there don't look like an acceptable doi code.
Lines 567-576: Suddenly the reference format changes to journal names and volumes in italics, and the year bold. Also, references 57 and 58 suddenly have web links, which are not given earlier. Be consistent throughout your references.
Author Response
Dear reviewer,
We would like to thank you for the comments and suggestions. Point-by-point responses to all the comments are provided below.
Comments and Suggestions for Authors: The authors present a detailed evaluation surface all-wave net radiation measurement from 240 sites with a good geographical and climatic region spread. They compare the measured data to the projections one would obtain from two major products commonly utilised for this purpose (GLASS and MERRA-2), and apply a variety of models to each of these.
Their results are credible and their findings useful. The work therefore deserves publication. The paper is also extremely well written, and does not require major corrections.
There are two points that I feel the authors should explain in more detail, alternatively develop further:
Reply: Thank you very much for your support. We appreciate the positive comments about the manuscript. We think your suggestions are also very important, here are some of our reply to your suggestions.
Point 1. The origin, nature and selection of the data is extremely unclear. While there is a map illustrating the locations used for the measurements, it is currently not possible to identify the sites with confidence. I am not suggesting a table with 240 entries, but rather some sort of a web link to enable an interested reader to access the list of sites. Or at least there needs to be an explanation about the criteria used to select the sites. If the study includes all sites for which such data exists (unlikely), then the paper should state that. Related to that, and perhaps even more important, there need to be details about how much data was used from each site, and how that data was selected (again, if every available data point between 2007 and 2017 was used, then the paper must state that). It says somewhere that the data used are daily averages. Is it possible that there is oversampling at particular times of the year, and under sampling at others? If one site has mainly winter data and another one mainly summer data, then that introduces a bias that may influence some of the regional or latitude dependences observed. I am not suggesting a repetition of the analysis on a seasonal basis, but rather to consider this effect in the evaluation of the results.
Reply: We appreciate the reviewer very much for the constructive comments and suggestions. We have made the following four revises according to your suggestion.
- We have added the web link that reader can download FLUXNET data. The revised sentence is provided as follows:
“The ground-measured data from 240 in-situ measurements were provided by AsiaFlux, AmeriFlux, LathuileFlux, the Asian Automatic Weather Station Network (ANN) Project, the Chinese Ecosystem Research Network (CERN) and individual principal investigators (PIs) of the FLUXNET website(https://fluxnet.org/).”
(2) We have listed the amount of in-situ measurements for different land cover types. The revised sentence is provided as follows:
“We acquired ground-measured data from 240 in-situ measurements including 34 cropland (CRO) sites, 6 deciduous needleleaf forest (DNF) sites, 28 deciduous broadleaf forest (DBF) sites, 16 evergreen broadleaf forest (EBF) sites, 64 evergreen needleleaf forest (ENF) sites, 10 savanna (SAW) sites, 14 shrubland (SHR) sites, 12 mixed forest (MF) sites and 56 grass and other types (GRA) sites between 2001 and 2009.”
(3) Our available in-situ measurements are between 2001 and 2009, and we have stated in our paper. The revised sentence are as follows:
“We acquired ground-measured data from 240 in-situ measurements including 34 cropland (CRO) sites, 6 deciduous needleleaf forest (DNF) sites, 28 deciduous broadleaf forest (DBF) sites, 16 evergreen broadleaf forest (EBF) sites, 64 evergreen needleleaf forest (ENF) sites, 10 savanna (SAW) sites, 14 shrubland (SHR) sites, 12 mixed forest (MF) sites and 56 grass and other types (GRA) sites between 2001 and 2009.”
- Thank you very much for your suggestions. We chose in-situ measurements of high quality, they would not have a lot of missing data for a long time. The original data were gap-filled using the MDS method. The processed data evenly distributed, and most sites don’t have mainly winter data or summer data. So, it should not have the problem of oversampling or under sampling.
Point 2. Related to the last sentence above, I think that the authors should go further in exploring the trends they observed in Section 3.2. Instead of just stating the regional peculiarities, offer some explanations for why these might be manifesting themselves. I don't feel that what is written in the paper at the moment goes far enough. For example, is it possible that the regional anomalities and trends over time are in any way related to drought/wet periods?
Reply: Thank you very much for your reminder. We have revised it according to your good suggestions. We have rewritten the sentences to explore the trends we observed. The paragraph is as following:
“Figure 4 shows the Rn trend of the two Rn products during 2007-2017. Rn decreased over 4.5% and 18.2% of global land area for GLASS and MERRA-2, respectively, and increased over 3.8% and 10.6% of global land area for MERRA-2 and GLASS. Generally, GLASS Rn exhibits increasing trend in most regions of global land area, and MERRA-2 Rn shows decreasing trend. The decreasing MERRA-2 Rn may indicate sparse vegetation and severe desertification occur in Central Africa. The different spatial patterns in Rn trends of the two Rn products may be caused by different model structure and input data of two Rn products [10, 34]. Figure 5 (a) presents the trends of globally averaged annual Rn of the two Rn products during the period 2007–2017. The globally averaged annual GLASS Rn exhibited insignificant upward trends. In contrast, MERRA-2 Rn showed insignificant decreasing trends. The two globally averaged annual Rn have small discrepancies that range from 0.3 W/m2 to 1.6 W/m2. Figure 5 (b) shows that the zonal means of two Rn products during the period from 2007–2017 changes with the latitude. Despite the similar spatial distributions in the two Rn products, MERRA-2 yielded higher Rn between latitudes of 55°S and 47°N, whereas GLASS had higher Rn in high latitude areas.”
Point 3. Line 26: I think "determined" is more accurate here than "detected".
Reply: Thanks. We revised the "detected" to "determined". The revised sentence is provided as follows:
“We then determined the resulting discrepancies in simulated annual global LE using SA model by merging five diagnostic LE models: RS-PM model, SW model, PT-JPL model, MS-PT model and SIM model.”
Point 4. Lines 28-30: I think leave out the acronyms "SA", "RS-PM", "PT-JPL", etc. Abstracts should be used for an overview, and not to define acronyms.
Reply: Thank you very much for your reminder. We revised those substantives to acronyms. The revised sentence is as follows:
“We then determined the resulting discrepancies in simulated annual global LE using a simple averaging model by merging five diagnostic LE models: RS-PM model, SW model, PT-JPL model, MS-PT model and SIM model.”
Point 5. Line 42: "CO2" should not be in italics.
Reply: Thanks for your reminder. We revised to co2
Point 6. Line 80: I guess you mean "Anderson et al.", and not "Martha".
Reply: Thanks for your reminder. We revised "Martha" to "Anderson et al."
Point 7. Line 101, 102 & 109: use a proper degree sign (as in line 107), not "o".
Reply: Thanks for your reminder. We revised to 0.05°.
Point 8. Equation 1 (and also at many other places in the paper): Please be consistent throughout when using italics (and when not using them). If "LE" (or any other of the short forms used in this paper, e.g.: Ta") is a variable parameter, then it would normally be in italics throughout the paper. If it is just an abbreviation, then it should not be in italics.
Reply: Thanks for your reminder. We have revised the variable parameters to normal format throughout the paper. And the equations been revised as following:
,
Point 9. Line 129: replace "acquired from this research" to "viewed elsewhere".
Reply: Thanks for your reminder. We revised "acquired from this research" to "viewed elsewhere". The revised sentence is provided as follows:
“More detail about the SW model can be viewed elsewhere [36]”
Point 10. Figs. 2 and 7: I personally think that these presently take up too much space, and I'm not sure if these plots are all necessary. To me they tell little more than that some climatic regions are far better sampled than others.
Reply: Thank you very much for your good suggestions. We have revised those figs that let them take up less space. However, we used scatter plot to display the relationship between estimation and ground-measured data. The different Land cover type has different scatter plot distribution. So, we think hold all scatter plot is more suitable.
Point 11. Lines 252-264: Don't you think that this would look much better in the form of a table?
Reply: Thank you very much for your good suggestions. We have revised the fig 6 to table 1. It was showed as follow:
|
Rn products |
Models |
R2 |
RMSE (W/m2) |
|
GLASS |
RS-PM |
0.56 |
28.8 |
|
SW |
0.53 |
31.7 |
|
|
PT-JPL |
0.6 |
27.4 |
|
|
MS-PT |
0.59 |
27.8 |
|
|
SIM |
0.59 |
27.2 |
|
|
SA |
0.6 |
26.6 |
|
|
MERRA-2 |
RS-PM |
0.5 |
33.1 |
|
SW |
0.49 |
34.7 |
|
|
PT-JPL |
0.49 |
34.2 |
|
|
MS-PT |
0.49 |
33.1 |
|
|
SIM |
0.48 |
33.2 |
|
|
SA |
0.52 |
31.8 |
Point 12. Figure 10: The horizontal ordinates -55, -25, 5, ... are rather odd. I think -60, -30, 0, 30, 60 would look far better (and is also more natural).
Reply: Thanks for your reminder. We revised “-55, -25, 5, ...” to “-60, -30, 0, 30, 60”. The revised Figure is showed as following:
Point 13. Line 413 and 505: I guess you mean "IEEE". Also check that the journal abbreviation used are the standard ones used by "Remote Sensing" journal.
Reply: Thanks for your reminder. We revised the “ieee” to “IEEE”. And we have checked the journal abbreviation. The revised sentence is as follows:
Xu, J.; Yao, Y.J.; Liang, S.L.; Liu, S.M.; Fisher, J.B.; Jia, K.; Zhang, X.T.; Lin, Y.; Zhang, L.L.; Chen, X.W. Merging the MODIS and Landsat Terrestrial Latent Heat Flux Products Using the Multiresolution Tree Method. IEEE T Geosci Remote 2019, 57, 2811-2823, doi:10.1109/Tgrs.2018.2877807.
Jiang, B.; Liang, S.L.; Jia, A.L.; Xu, J.L.; Zhang, X.T.; Xiao, Z.Q.; Zhao, X.; Jia, K.; Yao, Y.J. Validation of the Surface Daytime Net Radiation Product From Version 4.0 GLASS Product Suite. IEEE Geosci Remote S 2019, 16, 509-513, doi:10.1109/Lgrs.2018.2877625.
Point 14. Line 499: Clearly author Gao does not have the initials "HJIJoRS". The journal name is also missing.
Reply: Thanks for your reminder. We revised the citation to right format. The revised citation is as follow:
“Bisht, G.; Bras, R.L..; Sensing, R. Estimation of Net Radiation From the Moderate Resolution Imaging Spectroradiometer Over the Continental United States. IEEE T Geosci Remote 2011, 49, 2448-2462, doi: 10.1109/TGRS.2010.2096227.”
Point 15. Line 513: The characters given there don't look like an acceptable doi code.
Reply: Thanks for your reminder. We revised it to right doi code. The revised citation is showed as follow:
“PRIESTLEY, C.H.B.; TAYLOR, R.J. On the Assessment of Surface Heat Flux and Evaporation Using Large-Scale Parameters. Monthly Weather Review 1972, 100, 81-92, doi:https://doi.org/10.1175/1520-0493(1972)100<0081:OTAOSH>2.3.CO;2”
Point 16. Lines 567-576: Suddenly the reference format changes to journal names and volumes in italics, and the year bold. Also, references 57 and 58 suddenly have web links, which are not given earlier. Be consistent throughout your references.
Reply: Thanks for your reminder. We revised the format of those references. The revised citations are as follow:
“57. Yao, Y.; Liang, S.; Li, X.; Chen, J.; Liu, S.; Jia, K.; Zhang, X.; Xiao, Z.; Fisher, J.B.; Mu, Q., et al. Improving global terrestrial evapotranspiration estimation using support vector machine by integrating three process-based algorithms. Agricultural and Forest Meteorology 2017, 242, 55-74, doi: https://doi.org/10.1016/j.agrformet.2017.04.011.
- Yao, Y.; Liang, S.; Li, X.; Zhang, Y.; Chen, J.; Jia, K.; Zhang, X.; Fisher, J.B.; Wang, X.; Zhang, L., et al. Estimation of high-resolution terrestrial evapotranspiration from Landsat data using a simple Taylor skill fusion method. Journal of Hydrology 2017, 553, 508-526, doi: https://doi.org/10.1016/j.jhydrol.2017.08.013.
- Yao, Y.; Liang, S.; Yu, J.; Chen, J.; Liu, S.; Lin, Y.; Fisher, J.B.; McVicar, T.R.; Cheng, J.; Jia, K., et al. A simple temperature domain two-source model for estimating agricultural field surface energy fluxes from Landsat images. Journal of Geophysical Research: Atmospheres 2017, 122, 5211-5236, doi:10.1002/2016JD026370.”

Reviewer 2 Report
Summary:
“Discrepancies in the simulated global terrestrial latent heat flux from GLASS and MERRA-2 surface net radiation products” by Guo et al. uses net radiation from one reanalysis and one satellite product to calculate latent flux (LE) globally using the average of five model parameterizations. The results are then compared to 240 in situ surface observations from FLUXNET over a period of a decade. The study is straightforward and the presentation is generally clear. Results have the potential to provide useful guidance on global-scale parameterizations of LE. I am a little concerned that the scope is marginal for Remote Sensing being that the topic of remote sensing is incidental and the scope of the paper being more closely tied to modeling, but this is a decision for the editors. I also find one technical problem with the analysis that should be addressed, making a major revision required. While subjective, I also feel the study falls short of delivering on its potential and I hope that in a revision the authors can dig a little deeper to more completely interpret the results. My comments are as follows.
Major Comments:
- I was disappointed that the study stopped short of doing even basic sensitivity analyses to resolve some of the biases that were uncovered. Some examples:
- Do the biases in Rn reported in Section 3.1 explain the biases in LE? Or alternatively, are the errors in LE primarily associated with deficiencies in the parameterizations? I think you can easily test this by bias-correcting Rn, then recalculating LE and repeating the analysis in Section 3.3.
- How do the biases in LE relate to the assumed bias in the EC data from Eq. 2 and 3?
- Some additional analysis of the individual models and identification of the biggest sensitivities in the parameterizations. What I mean is that (ref to Fig 6), the differences between the parameterizations is smaller than the differences between MERRA and GLASS, which supports the hypothesis that Rn is very important (shared input between the parameterizations), and the unique aspects of the parametrizations (e.g., inclusion of NDVI) are minor. However, it does not help me understand how big of a factor uncertainty in Rn is compared to uncertainty in other shared inputs, like T and RH.
- There is a zonal artifact in the GLASS data near 50N (appearing for example in Figs 3a, 4a, 5b). You maybe acknowledge this around L229. Perhaps this is a bug in your processing code or maybe this is an issue native to GLASS? Regardless, it is non-physical and absolutely must be resolved because it impacts the principle results you are reporting. It seems to propagate into the LE values too, although less discernable (Figs. 8-10).
Minor Comments:
First, two general comments then some specific ones.
- Please make sure there are x- and y-axes labels in all figures.
- The term “Rn products” and similar that is used throughout is very confusing because Rn means “net radiation”, but the term “Rn products” is sometimes meant to refer to presentation of the variable latent heat derived from the Rn of GLASS and MERRA.
Introduction:
- Consider restructuring the intro a little. Some of the content of the second to last paragraph nicely frames the problem this study is addressing and should maybe appear earlier.
- Can you provide a little more introduction to the five diagnostic LE models?
- You list a number of products but are only choosing to compare GLASS and MERRA. Why these? Why not the others as well?
Section 2.2:
I don’t think you need Eq. 1. Just say you averaged the models and reference [18]. Make this statement after you introduce the models and just make this one section, 2.2. Also, we need more of a primer on the diagnostic models. For example, in 2.2.2(1) we jump right into describing the changes to the diagnostic model from prior versions without really knowing yet how the diagnostic models are structured. One framework for the introductory material I want to see is to develop it around Rn. The 5 models you describe have a number of shared and unique input variables. They all use Rn (please specify if this is true for model (1) as well) so presumably that is one reason why you focus on Rn. But there are others as well (like RH). Why is Rn the most important input to test?
Section 2.2.3:
I found this section confusing. E.g, FPAR/LAI? What do you mean “to similar LE with two Rn products”?
Section 3.1:
- It would be helpful to include a table somewhere for the reader to reference the acronyms of the land-cover types. Else write the land cover types out. You can probably just modify Fig 1 to spell out the types there alongside the acronyms.
- At line 184 you say you are comparing Rn from GLASS/MERRA to “EC” observations, which is not true. EC is eddy covariance and refers to Hs and Hl turbulent fluxes. You are comparing the radiative fluxes here, which may be available from FLUXNET, but are measured using radiometers, not sonic anemometers and gas analyzers. Since you use the FLUXNET radiometric data, you should describe it in the Data section.
- There do seem to be some fairly consistent biases in Fig 2 with the largest value being the observations, the lowest being GLASS and MERRA being in between, but both being lower than observations. Do you have any interpretation for the meaning of this? Is it in the longwave or the shortwave?
Section 3.2:
Lines 219-221 (sentence begin “Rn significantly….”) is nonsensical.
Section 3.3:
Lines 265-267: This is awkwardly stated, but I think you mean that the CRO sites have unnaturally-high LE that would not be accounted for by the models because of contributions from nearby irrigation.
Section 4-5:
I think you can just combine these into one section, “Discussion and Conclusions” since Section 4 is already a mixture.
Technical Editing Comments:
Line 112: The “A” in SA probably standard for “averaging” so shouldn’t it be “simple model averaging (SA)”?
Line 119: PM = Penman Montieth?
Lines 237-239: Rephrase “It can be noticed”, which is awkward to read.
Author Response
Dear reviewer,
We would like to thank you for the comments and suggestions. Point-by-point responses to all the comments are provided below.
Comments and Suggestions for Authors: “Discrepancies in the simulated global terrestrial latent heat flux from GLASS and MERRA-2 surface net radiation products” by Guo et al. uses net radiation from one reanalysis and one satellite product to calculate latent flux (LE) globally using the average of five model parameterizations. The results are then compared to 240 in situ surface observations from FLUXNET over a period of a decade. The study is straightforward and the presentation is generally clear. Results have the potential to provide useful guidance on global-scale parameterizations of LE. I am a little concerned that the scope is marginal for Remote Sensing being that the topic of remote sensing is incidental and the scope of the paper being more closely tied to modelling, but this is a decision for the editors. I also find one technical problem with the analysis that should be addressed, making a major revision required. While subjective, I also feel the study falls short of delivering on its potential and I hope that in a revision the authors can dig a little deeper to more completely interpret the results. My comments are as follows.
Major Comments:
I was disappointed that the study stopped short of doing even basic sensitivity analyses to resolve some of the biases that were uncovered. Some examples:
Point 1: Do the biases in Rn reported in Section 3.1 explain the biases in LE? Or alternatively, are the errors in LE primarily associated with deficiencies in the parameterizations? I think you can easily test this by bias-correcting Rn, then recalculating LE and repeating the analysis in Section 3.3.
Reply: Thank you very much for your suggestions. Your opinion is precious to us. In fact, there are many parameters associated with accuracy of LE, and Rn influence LE can be tested by bias-correcting, and then recalculating LE. However, the mainly purpose of this paper is research how different Rn product influence the LE retrievals. We think compare differently Rn product directly instead of bias-correcting is more suitable for our purpose. In our following study how Rn (not Rn product) influence LE retrievals, we pleased to use bias-correcting.
To research how Rn product, influence LE retrievals, we compare two different Rn product (Rn product GLASS that based on remote sensing data and Rn product MERRA-2 that based on meteorological data assimilation) firstly. Then we analysis the accuracy of LE based on two Rn product at in-situ measurements and global scale. They all proved that LE based-on GLASS Rn is more accuracy than based on MERRA-2 Rn. So, we think GLASS Rn product is more suitable for LE retrievals.
Point 2: How do the biases in LE relate to the assumed bias in the EC data from Eq. 2 and 3?
Reply: Thanks, the energy (Rn, LE, H and G) that measured by EC is imbalance.
The models were developed based on surface energy balance. Therefore, we used Eq. 2 and 3 to correct LE. In fact, the bias of corrected LE is related to original LE. We plotted a scatter plot to show the relationship between original LE and corrected LE.
Point 3: Some additional analysis of the individual models and identification of the biggest sensitivities in the parameterizations. What I mean is that (ref to Fig 6), the differences between the parameterizations is smaller than the differences between MERRA and GLASS, which supports the hypothesis that Rn is very important (shared input between the parameterizations), and the unique aspects of the parametrizations (e.g., inclusion of NDVI) are minor. However, it does not help me understand how big of a factor uncertainty in Rn is compared to uncertainty in other shared inputs, like T and RH.
Reply: Thank you very much for your suggestions. As you said, the other inputs, like Ta and RH, are also important to LE estimation. But our objective in this study is to assess the influence of different Rn products on LE.
Therefore, we used the same inputs (except Rn) and the same model to yield GLASS-based LE and MERRA-2-based LE. The only difference of LE estimation is Rn input. The two Rn products were simulated LE with same other input (e.g. Ta, RH, LAI) and model. Thus, the difference of LE only came from the Rn products.
Point 4: There is a zonal artifact in the GLASS data near 50N (appearing for example in Figs 3a, 4a, 5b). You maybe acknowledge this around L229. Perhaps this is a bug in your processing code or maybe this is an issue native to GLASS? Regardless, it is non-physical and absolutely must be resolved because it impacts the principle results you are reporting. It seems to propagate into the LE values too, although less discernable (Figs. 8-10).
Reply: Thank you very much for your reminder. Your opinion is precious to us. This is an issue native to GLASS Rn. GLASS Rn was produced with remote sensing data. It has missing data around the north pole several times a year (most in the spring and winter). Although, the missing data was temporally interpolating, it had a problem of visual representation. We have revised the description of GLASS Rn; the paragraph is showed as follows:
“The Global LAnd Surface Satellite (GLASS) beta version Rn was generated using the multivariate adaptive regression splines (MARS) method, Moderate-Resolution Imaging Spectroradiometer (MODIS) data and MERRA-2 meteorological reanalysis data [10]. The MARS model of GLASS Rn was trained by FLUXNET ground-measured Rn data and their corresponding remote sensing data and reanalysis MERRA-2 data[10]. Daily GLASS Rn is a perennial remote sensing product with spatial resolution of 0.05°, beginning in 2000. This Rn product not only has better spatial resolution (0.05°) and temporal resolution (daily) but also has higher accuracy than reanalysis data[34]. We therefore used the GLASS Rn product as forcing data of the LE model. The GLASS Rn has missing data around the north pole several times a year (most in the spring and winter) because it was produced with remote sensing data. The missing data was temporally filled using the algorithm described by Zhao et al. [35].”
Minor Comments:
First, two general comments then some specific ones.
Point 5: Please make sure there are x- and y-axes labels in all figures.
Reply: Thanks. We have added the x- and y-axes labels in all figures. The revised Figure is showed as following:
Point 6: The term “Rn products” and similar that is used throughout is very confusing because Rn means “net radiation”, but the term “Rn products” is sometimes meant to refer to presentation of the variable latent heat derived from the Rn of GLASS and MERRA.
Reply: Thanks. In this paper, we used the “Rn products” to represent GLASS Rn and MERRA-2 Rn. We used “LE based-on GLASS Rn product”, “LE based-on MERRA-2 Rn product”, to represent LE derived from the Rn of GLASS and MERRA-2, respectively.
Point 7: Consider restructuring the intro a little. Some of the content of the second to last paragraph nicely frames the problem this study is addressing and should maybe appear earlier.
Reply: Thank you very much for your suggestions. You advise is very constructive. Although the introduction can be restructured by discussing the addressed problem earlier. However, this paper is about LE estimation using Rn products, we think it is more suitable to discuss LE and Rn firstly. So, in introduction, the first and second paragraphs are description about LE and Rn. The third and fourth paragraphs are about Rn is an important input to LE models and there are different kinds Rn products. Those paragraphs are the background. If the second to last paragraph appear earlier, the reader may be confused the influence of Rn to LE.
Point 8: Can you provide a little more introduction to the five diagnostic LE models?
Reply: Thank you very much for your suggestions. We have added an appendix to introduce the five diagnostic LE models. The appendix is showed as follows:
“Mu et al.[7] developed the remote sensing-based Penman-Monteith model by modifying Penman-Monteith logic. The RS-PM can be calculated as follows:
|
, |
(A1) |
Where D is the slope of the relationship of saturation vapour pressure and air temperature, ρ is the air density, g is the psychrometric constant, Cp is the specific heat of the air, VPD is the vapour pressure deficit, rs is the surface resistances and ra is the aerodynamic resistances.
The SW model divides LE into soil evaporation and vegetation transpiration[36], the model can be written as:
|
, |
(A2) |
|
, |
(A3) |
|
, |
(A4) |
|
, |
(A5) |
|
, |
(A6) |
|
, |
(A7) |
Where LEs and LEa are the soil evaporation and vegetation transpiration, raa is the aerodynamic resistances from reference highness to vegetation highness, rac is the aerodynamic resistances from foliage to canopy highness, ras is the aerodynamic resistances from surface to canopy highness, rss is the surface resistance of soil and rsc is the surface resistance of vegetation.
The PT-JPL LE model was proposed by Fisher on the basis of the Priestley–Taylor logic[37], it can be calculated as follows:
|
, |
(A8) |
|
, |
(A9) |
|
, |
(A10) |
|
, |
(A11) |
Where LES, LEc and LEi are soil evaporation, vegetation transpiration and evaporation of canopy interception. fwet, fsm and fg are the surface wet fraction (RH4), moisture constraint of soil (RHVPD) and canopy fraction (fAPAR/fIPAR). fT and fM are the constraint of temperature and constraint of moisture. Rns and Rnc are Rn of soil and Rn of vegetation.
The MS-PT model was developed by Yao et al. [15] and it can be written as:
|
, |
(A12) |
|
, |
(A13) |
|
, |
(A14) |
|
, |
(A15) |
|
, |
(A16) |
|
, |
(A17) |
|
, |
(A18) |
Where LEds, LEws, LEv and LEic are the soil evaporation, wet soil evaporation, vegetation transpiration and interception evaporation. NDVImin and NDVImax are 0.05 and 0.95[15].
Wang et al. developed the SIM model [38], it can be calculated as follows:
|
, |
(A19) |
|
“ |
|
Point 9: You list a number of products but are only choosing to compare GLASS and MERRA. Why these? Why not the others as well?
Reply: Thanks. We chose GLASS and MERRA-2 Rn product mainly because they are different type Rn product (one based on remote sensing and one based on Meteorological data assimilation).
GLASS Rn is a long-term remote sensing product with spatial resolution of 0.05°, beginning in 2000. This Rn product has not only high spatial resolution (0.05°) but high temporal resolution (daily).
MERRA-2 Rn covers a long-term time span and is spatially and temporally continuous with no missing data. It is one of most popular reanalysis data.
We have produced GLASS ET product using MERRA Rn and other inputs; and it has pretty good performance. We want to see if GLASS Rn will improve the accuracy of the GLASS ET product. We will compare more Rn products in the future.
Point 10: I don’t think you need Eq. 1. Just say you averaged the models and reference [18]. Make this statement after you introduce the models and just make this one section, 2.2.
Reply: Thanks. We have removed the Eq. 1. And we make this one section. The revised sentence is showed as follow:
“2.2.1 LE models
We used the five diagnostic LE models to simulate LE and the description of the five diagnostic LE models was demonstrated in Appendix A.
(1) RS-PM model. The remote sensing-based Penman-Monteith (RS-PM) model was modified from the MODIS global LE model [14]. Mu et al. [7] deigned the model by (1) replacing vegetable cover fraction with Fraction of Absorbed Photosynthetically Active Radiation (FPAR), (2) adding night-time LE, (3) estimating soil heat flux, (4) developing estimates of canopy resistance, aerodynamics and boundary-level, (5) dividing LE into interception evaporation, canopy transpiration, soil evaporation and wet soil evaporation. Rn, RH, Ta, water pressure (e) and LAI were required to drive the model.
(2) SW model. The Shuttleworth–Wallace dual-source (SW) model divided LE into soil evaporation and vegetation transpiration. Each component of SW-based LE was calculated by the Penman–Monteith algorithm. The SW model assumes aerodynamic mixing arising at a mean canopy source within the canopy. More detail about the SW model can be viewed elsewhere [36]. The SW model requires Rn, RH, Ta, e, wind speed and LAI.
(3) PT-JPL model. The Priestley-Taylor of the Jet Propulsion Laboratory (PT-JPL) LE model was proposed by Fisher on the basis of the Priestley–Taylor model [37]. Fisher et al. modified the Priestley–Taylor model using the atmosphere and ecophysiology to calculate the actual LE. The input forcing data to generate PT-JPL LE data is Rn, RH, Ta, e, LAI and FPAR.
(4) MS-PT model. The modified satellite-based PT (MS-PT) model was designed by Yao et al. and is based on the PT-JPL model [15]. Yao et al. used the diurnal temperature range (DT) to calculate the apparent thermal inertia (ATI) that represents soil moisture constraints. The MS-PT model divides LE into four components: unsaturated surface soil evaporation, saturated surface soil evaporation, vegetation canopy transpiration and vegetation interception evaporation. Since the MS-PT model reduces the parameters of PT-JPL, it only needs Rn, Ta, DT NDVI as inputs.
(5) SIM model. The simple hybrid LE (SIM) model was designed by Wang et al. (2008) by considering the influence of soil moisture on the LE parameterization [38]. This model introduces the influence of soil moisture to the LE parameterization. The coefficients of this model were calibrated using LE measurements in America from 2002 to 2005. The input variables of the SIM model are Rn, Ta, DT and NDVI.
To examine the effects of Rn on estimated LE, we used the simple model averaging (SA) method to merge five LE models for estimating terrestrial LE. Previous studies pointed out that the SA method performs better than individual models [18]. The SA method calculates terrestrial LE by averaging each single LE model.”
Point 11: Also, we need more of a primer on the diagnostic models. For example, in 2.2.2(1) we jump right into describing the changes to the diagnostic model from prior versions without really knowing yet how the diagnostic models are structured. One framework for the introductory material I want to see is to develop it around Rn.
Reply: Thanks. We have added an appendix to introduce the five diagnostic LE models. The appendix descripts the structure of diagnostic models. The appendix has been placed above.
Point 12: The 5 models you describe have a number of shared and unique input variables. They all use Rn (please specify if this is true for model (1) as well) so presumably that is one reason why you focus on Rn. But there are others as well (like RH). Why is Rn the most important input to test?
Reply: Thanks. Rn is the energy source for LE, and it has a great influence on estimation LE. According to Terrestrial Surface Energy Balance model, Rn =LE+ H + G (H is sensible heat flux, G is soil heat flux), Rn controls the energy exchange between the terrestrial ecosystem and the atmosphere, and it has a great influence on LE.
There are many studies that conclude Rn has critical influence on LE estimation, like:
Wang K , Wang P , Li Z , et al. A simple method to estimate actual evapotranspiration from a combination of net radiation, vegetation index, and temperature[J]. Journal of Geophysical Research: Atmospheres, 2007, 112(D15).
Wang, K., and S. Liang, 2008: An Improved Method for Estimating Global Evapotranspiration Based on Satellite Determination of Surface Net Radiation, Vegetation Index, Temperature, and Soil Moisture. J. Hydrometeor., 9, 712–727, https://doi.org/10.1175/2007JHM911.1.
Hasler, N., and R. Avissar, 2007: What Controls Evapotranspiration in the Amazon Basin?. J. Hydrometeor., 8, 380–395, https://doi.org/10.1175/JHM587.1.
The other inputs are also important inputs, we will inspect influence of other inputs to LE estimation in the future.
Point 13: I found this section confusing. E.g, FPAR/LAI? What do you mean “to similar LE with two Rn products”?
Reply: Thanks. We have revised the abbreviation to full name. My sentence is “to simulate LE with two Rn products”, my meaning was we used the two Rn products to simulate LE. The revised sentence is as follow:
“To simulate LE with two Rn products, we used the MODIS 8-day Fraction of Photosynthetically Active Radiation (FPAR) and Leaf Area Index (LAI) product [39] with a spatial resolution of 1 km, and the 8-day average FPAR/LAI was temporally interpolated to daily FPAR/LAI values using linear interpolation. Additionally, the 16-day MODIS Normalized Difference Vegetation Index (NDVI) and Enhanced Vegetation Index (EVI) [40] was adopted to drive the LE models.”
Point 14: It would be helpful to include a table somewhere for the reader to reference the acronyms of the land-cover types. Else write the land cover types out. You can probably just modify Fig 1 to spell out the types there alongside the acronyms.
Reply: Thanks. We have written the full name of land-cover types.
“We acquired ground-measured data from 240 in-situ measurements including 34 cropland (CRO) sites, 6 deciduous needleleaf forest (DNF) sites, 28 deciduous broadleaf forest (DBF) sites, 16 evergreen broadleaf forest (EBF) sites, 64 evergreen needleleaf forest (ENF) sites, 10 savanna (SAW) sites, 14 shrubland (SHR) sites, 12 mixed forest (MF) sites and 56 grass and other types (GRA) sites between 2001 and 2009.”
Point 15: At line 184 you say you are comparing Rn from GLASS/MERRA to “EC” observations, which is not true. EC is eddy covariance and refers to Hs and Hl turbulent fluxes. You are comparing the radiative fluxes here, which may be available from FLUXNET, but are measured using radiometers, not sonic anemometers and gas analyzers. Since you use the FLUXNET radiometric data, you should describe it in the Data section.
Reply: Thanks. We have revised this mistake. And we also revised the Data section. The revised sentences are as follow:
“Rn products and LE models were validated and evaluated using ground-measured data from in-situ radiation measurements and eddy covariance flux tower sites.”
“To evaluate the quality of the two Rn products, we used ground-measured data from 240 in-situ radiation measurements.”
Point 16: There do seem to be some fairly consistent biases in Fig 2 with the largest value being the observations, the lowest being GLASS and MERRA being in between, but both being lower than observations. Do you have any interpretation for the meaning of this? Is it in the longwave or the shortwave?
Reply: Thanks. The MERRA Rn higher than observations the MERRA-2 Rn is overestimated in some land cover types (e.g. EBF, GRA…). However, GLASS Rn is underestimated all land cover type. The GLASS Rn adopted machine learning method to train ground measured Rn, corresponding remote sensing data and reanalysis MERRA-2 data. The performance of machine learning is mainly depending on train data. The underestimation phenomenon of GLASS Rn may result from the train data.
Point 17: Lines 219-221 (sentence begin “Rn significantly….”) is nonsensical.
Reply: Thanks. We have revised the sentence as follow.
“Rn decreased over 4.5% and 18.2% of global land area for GLASS and MERRA-2, respectively, and increased over 3.8% and 10.6% of global land area for MERRA-2 and GLASS.”
Point 18: Lines 265-267: This is awkwardly stated, but I think you mean that the CRO sites have unnaturally-high LE that would not be accounted for by the models because of contributions from nearby irrigation.
Reply: Thanks. We have revised the sentence as follow.
“For the CRO sites, the two products underestimate the values of LE, since they do not consider the influence of irrigation on LE.”
Point 19: I think you can just combine these into one section, “Discussion and Conclusions” since Section 4 is already a mixture.
Reply: Thank you very much for your suggestion. Your suggestion is very constructive. However, the Remote Sensing needs section of “Discussion” and section of “Conclusions”.
Point 20: Line 112: The “A” in SA probably standard for “averaging” so shouldn’t it be “simple model averaging (SA)”?
Reply: Thank you very much for your reminder. We have revised “simple model (SA) averaging” to “simple model averaging (SA)”
Point 20: Line 119: PM = Penman Monteith?
Reply: Thank you very much for your reminder. We have revised “PM” to “Penman Monteith”.
Point 21: Lines 237-239: Rephrase “It can be noticed”, which is awkward to read.
Reply: Thank you very much for your suggestion. We have revised the sentence as follow.
“The GLASS-based LE have higher R2 and lower RMSE compared to the MERRA-2-based LE for different LE models (R2 increased by 0.04~0.11 and RMSE decreased by 3~6.8 W/m2).”

Reviewer 3 Report
Proposed manuscript analyses accuracy/discrepancies of daily net radiation (Rn) products from GLASS and MERRA-2 with ground based Rn measurement. Comparison is based on data from 240 eddy covariance tower sites of FLUXNET network. The effects of Rn discrepancies were analysed on comparison of six models of latent het flux (LE) calculated with using GLASS and MERRA-2 Rn products with eddy covariance LE measurement. I think the manuscript covers very important topic. The results can lead to improvement of produts of both compared systems as well as it can be heplfull for users of these products.
I have some comments on the manuscript, mostly on methodology. The methodology of ground measurement Rn and LE data and accuracy of these data should be described more in detail. The SA method has been calculated from five models used in the work. I think a usage of mean of the methods as diagnostic method (SA) could be a bit problematic because a result can depends on an accuracy of these methods. Would be a result better if the weakest method (here SW method) will be removed from SA calculation? Please discuss it. I recommend to add a more detailed description of LE models to appendix of the paper.
An effect of topography (shape of surface) on net radiation should be discussed, in my opinion – Rn is usually measured on horizontal surface. An effect of land cover variability within pixels area of GLASS and MERRA-2 on Rn and LE should be discussed more in detail.
Please, add a description and units to axes of graphs (fig. 2 and 7) and units to colour scales at figs 3, 4, 8 and 9.
Author Response
Dear reviewer,
We would like to thank you for the comments and suggestions. Point-by-point responses to all the comments are provided below.
Comments and Suggestions for Authors: Proposed manuscript analyses accuracy/discrepancies of daily net radiation (Rn) products from GLASS and MERRA-2 with ground-based Rn measurement. Comparison is based on data from 240 eddy covariance tower sites of FLUXNET network. The effects of Rn discrepancies were analysed on comparison of six models of latent het flux (LE) calculated with using GLASS and MERRA-2 Rn products with eddy covariance LE measurement. I think the manuscript covers very important topic. The results can lead to improvement of products of both compared systems as well as it can be helpful for users of these products.
Reply: Thank you very much for your support. We appreciate the positive comments about the manuscript. We think your suggestions are also very important, here are some of our responses to your suggestions.
Point 1: The methodology of ground measurement Rn and LE data and accuracy of these data should be described more in detail.
Reply: Thanks. We have added web link that can download at in-situ measurement Flux net data, and described the Rn data that has been used in detail. And we have descripted the accuracy of LE and Rn. The revised version is showed as follow:
“Rn products and LE models were validated and assessed using ground-measured data from in- situ radiation measurements and eddy covariance flux tower sites. The ground-measured data from 240 in-situ measurements were provided by AsiaFlux, AmeriFlux, LathuileFlux, the Asian Automatic Weather Station Network (ANN) Project, the Chinese Ecosystem Research Network (CERN) and individual principal investigators (PIs) of the FLUXNET website(https://fluxnet.org/). The in-situ measurements are mostly mainly distributed located in North America, Europe and Asia (Figure 1). We acquired ground-measured data from 240 in-situ measurements including 34 cropland (CRO) sites, 6 deciduous needleleaf forest (DNF) sites, 28 deciduous broadleaf forest (DBF) sites, 16 evergreen broadleaf forest (EBF) sites, 64 evergreen needleleaf forest (ENF) sites, 10 savanna (SAW) sites, 14 shrubland (SHR) sites, 12 mixed forest (MF) sites and 56 grass and other types (GRA) sites between 2007 and 2009. The LE and Rn from ground-measured data can have an error of 10% [42,61]”
Point 2: I think a usage of mean of the methods as diagnostic method (SA) could be a bit problematic because a result can depends on an accuracy of these methods. Would be a result better if the weakest method (here SW method) will be removed from SA calculation?
Reply: Thank you very much for your suggestion. As you known, the SW had a weakness when we used whole dataset. If we remove SW model, the accuracy of SA model is not good when we used whole dataset. However, the SW model had pretty good performance in several land cover types (e.g. DBF). We remove it from SA calculation, the accuracy of this land cover types (e.g. DBF) will decrease.
Point 3: I recommend to add a more detailed description of LE models to appendix of the paper.
Reply: Thank you very much for your reminder. We have added a more detailed description of LE models to appendix of the paper. The appendix is showed as follows:
Mu et al.[7] developed the remote sensing-based Penman-Monteith model by modifying Penman-Monteith logic. The RS-PM can be calculated as follows:
|
, |
(A1) |
Where D is the slope of the relationship of saturation vapour pressure and air temperature, ρ is the air density, g is the psychrometric constant, Cp is the specific heat of the air, VPD is the vapour pressure deficit, rs is the surface resistances and ra is the aerodynamic resistances.
The SW model divides LE into soil evaporation and vegetation transpiration[36], the model can be written as:
|
, |
(A2) |
|
, |
(A3) |
|
, |
(A4) |
|
, |
(A5) |
|
, |
(A6) |
|
, |
(A7) |
Where LEs and LEa are the soil evaporation and vegetation transpiration, raa is the aerodynamic resistances from reference highness to vegetation highness, rac is the aerodynamic resistances from foliage to canopy highness, ras is the aerodynamic resistances from surface to canopy highness, rss is the surface resistance of soil and rsc is the surface resistance of vegetation.
The PT-JPL LE model was proposed by Fisher on the basis of the Priestley–Taylor logic[37], it can be calculated as follows:
|
, |
(A8) |
|
, |
(A9) |
|
, |
(A10) |
|
, |
(A11) |
Where LES, LEc and LEi are soil evaporation, vegetation transpiration and evaporation of canopy interception. fwet, fsm and fg are the surface wet fraction (RH4), moisture constraint of soil (RHVPD) and canopy fraction (fAPAR/fIPAR). fT and fM are the constraint of temperature and constraint of moisture. Rns and Rnc are Rn of soil and Rn of vegetation.
The MS-PT model was developed by Yao et al. [15] and it can be written as:
|
, |
(A12) |
|
, |
(A13) |
|
, |
(A14) |
|
, |
(A15) |
|
, |
(A16) |
|
, |
(A17) |
|
, |
(A18) |
Where LEds, LEws, LEv and LEic are the soil evaporation, wet soil evaporation, vegetation transpiration and interception evaporation. NDVImin and NDVImax are 0.05 and 0.95[15].
Wang et al. developed the SIM model [38], it can be calculated as follows:
|
, |
(A19) |
Point 4: An effect of topography (shape of surface) on net radiation should be discussed, in my opinion – Rn is usually measured on horizontal surface.
Reply: Thank you very much for your reminder. topography does affect the accuracy of Rn, and we have discussed it in the discussion part the revised sentences are showed as follows:
“The topography (e.g. aspect, slope, elevation,) is also a critical factor for Rn simulation. The accurately topographical information can help us estimate reasonable direct solar radiation [60]. Both the GLASS and MERRA-2 Rn products do not consider the influence of topography on Rn. We will use topographical information to improve Rn estimation in the future.”
Point 5: An effect of land cover variability within pixels area of GLASS and MERRA-2 on Rn and LE should be discussed more in detail.
Reply: Thank you very much for your reminder. We have discussed how land cover variability influence Rn and LE in detail.
the revised sentences are showed as follows:
“The accuracy of Rn products varies for diversity of land cover types because the differences in surface biophysical parameters (e.g. surface albedo, surface emissivity, soil moisture) across multiple biomes alter surface energy budget. “
“For different land cover types, the discrepancies in LE estimation vary greatly due to the difference of surface moisture status and vegetation stomatal conductance. “
Point 6: Please, add a description and units to axes of graphs (fig. 2 and 7) and units to colour scales at figs 3, 4, 8 and 9.
Reply: Thank you very much for your reminder. We have added description and units to axes of graphs (fig. 2 and 7). And we have added units at figs 3, 4, 8 and 9. The revised figure are showed as following:
fig. 2
fig.7
figs 3, 4, 8 and 9

Round 2
Reviewer 3 Report
I have no more coments on the manuscript.
Author Response
thanks